# Combination of T cell-redirecting bispecific antibody ERY974 and chemotherapy reciprocally enhances efficacy against non-inflamed tumours

Yuji Sano [1] ✉, Yumiko Azuma[1], Toshiaki Tsunenari[1], Yoko Kayukawa[1], Junko Shinozuka[2], Etsuko Fujii[2], Jun Amano[2], Yukari Nishito[1], Toru Maruyama[1], Yasuko Kinoshita[1], Yuichiro Sakamoto[2], Ayae Yoshida[3], Yoko Miyazaki[1], Yuta Sato[2], Chifumi Teramoto-Seida[2], Takahiro Ishiguro[4], Takayoshi Tanaka[4], Takehisa Kitazawa[1] & Mika Endo[4]

Identifying a strategy with strong efficacy against non-inflamed tumours is vital in cancer immune therapy. ERY974 is a humanized IgG4 bispecific T cell-redirecting antibody that recognizes glypican-3 and CD3. Here we examine the combination effect of ERY974 and chemotherapy (paclitaxel, cisplatin, and capecitabine) in the treatment of non-inflamed tumours in a xenograft model. ERY974 monotherapy shows a minor antitumour effect on non-inflamed NCI-H446 xenografted tumours, as infiltration of ERY974-redirected T cells is limited to the tumour-stromal boundary. However, combination therapy improves efficacy by promoting T cell infiltration into the tumour centre, and increasing ERY974 distribution in the tumour. ERY974 increases capecitabine-induced cytotoxicity by promoting capecitabine conversion to its active form by inducing thymidine phosphorylase expression in non-inflamed MKN45 tumour through ERY974-induced IFNγ and TNFα in T cells. We show that ERY974 with chemotherapy synergistically and reciprocally increases anti-tumour efficacy, eradicating non-inflamed tumours.

Immune checkpoint inhibitors (ICIs), such as anti-CTLA4, anti-PD-1 and anti-PD-L1 antibodies, show promising efficacy in a wide range of cancer types and represent a breakthrough in cancer therapy[1]. However, with a response rate of approximately 30%, numerous patients do not benefit from ICI therapy[2]. Moreover, ICIs show strong efficacy in inflamed tumours, where the number of T cells is sufficient, but not in non-inflamed tumours where T cells are scarce[3]. As a result, strategies for converting a non-inflamed tumour microenvironment (TME) to an inflamed one have gained increasing attention in the field of cancer immunotherapy[4].

T cell-redirecting antibody (TRAB), one of the most promising novel modalities for creating an inflamed TME, can achieve efficacy by exploiting the strong cytotoxic activity of T cells; several TRABs are currently being evaluated in clinical trials[5]. For example, teben-tafusp, an atypical TRAB consisting of an anti-CD3 arm and a high-affinity T-cell receptor (TCR) that recognises gp100 antigen, which is

[1]Research Division, Chugai Pharmaceutical Co., Ltd., 200 Kajiwara, Kamakura, Kanagawa 247-8530, Japan. [2]Research Division, Chugai Pharmaceutical Co., Ltd., 1-135 Komakado, Gotemba, Shizuoka 412-8513, Japan. [3]Chugai Research Institute for Medical Science, Inc., 1-135 Komakado, Gotemba, Shizuoka 412-8513, Japan. [4]Translational Research Division, Chugai Pharmaceutical Co., Ltd., 1-1Nihonbashi-Muromachi 2-Chome Chuo-ku, Tokyo 103-8324, Japan. ✉e-mail: sanoyuj@chugai-pharm.co.jp

presented to the TCR through major histocompatibility complex (MHC)[6], showed promising efficacy in phase III clinical trials for patients with uveal melanoma[7]. Furthermore, AMG160, a different type of TRAB known as a bispecific T-cell engager (BiTE) that targets PSMA, showed promising efficacy in patients with metastatic castration-resistant prostate cancer[8], and TNB 585, another PSMA-targeted TRAB[9], is also in phase I trials. TRAB recognises a tumour antigen with one arm and a T cell with the other arm, cross-linking them to activate the T cells. T cells activated by TRAB secrete cytokines, chemokines, or other molecules required for cytotoxicity, such as granzyme and perforin, to attack tumour cells[10,11]. Moreover, TRAB can promote T cell proliferation at the tumour site. As an application of this concept, CEA-TCB, a bispecific antibody targeting carcinoembryonic antigen (CEA) and CD3, currently in clinical development both as a monotherapy and combination therapy with atezolizumab, showed efficacy in patients with microsatellite stable metastatic colorectal cancer[12], in which cytotoxic T cells are scarce due to the low number of neoantigens and where ICI monotherapy shows less efficacy[13,14].

ERY974, a TRAB that targets glypican-3 (GPC3) and CD3[15] currently undergoing evaluation in a clinical trial (JapicCTI-194805/NCT05022927), has a mutated IgG4-Fc, which loses binding to Fc gamma receptors but still binds to the neonatal Fc receptor. Therefore, antigen-independent cytokine release syndrome could be mitigated, and a weekly intravenous injection is enough to maintain a sufficient level of ERY974 in the blood, as per our previous preclinical data[15]. Notably, ERY974 has shown significant efficacy against non-inflamed LLC1/hGPC3 tumours, where ICIs failed to show efficacy[15]. GPC3 is an ideal target for TRAB because its expression in normal tissues is very low[16], unlike other TRAB tumour antigens. Moreover, GPC3 is widely expressed in various tumour types, including liver[17], lung[18], gastric[19], head and neck[20] and ovarian cancers[21].

Herein, we demonstrate that ERY974 monotherapy for non-inflamed tumours is efficacious, though not sufficiently to achieve tumour regression, by employing NCI-H446 and MKN45 xenograft tumour models. We find that the combination of ERY974 and chemotherapy is a suitable strategy for improving ERY974 efficacy in such non-inflamed tumours. Our findings provide insights and guidance into strategies for clinical trials using ERY974.

## Results

### ERY974 shows minor antitumour efficacy in non-inflamed tumours

Although GPC3 expression is the critical biomarker that determines the efficacy of ERY974 as we have previously shown[15], it remains unclear whether GPC3 expression is sufficient to predict the efficacy of ERY974. To examine whether GPC3 expression is the sole biomarker for determining the efficacy of ERY974, we compared ERY974 efficacy between various tumour models with comparable GPC3 expression levels. We selected PC10 and NCI-H446 cells to represent high GPC3-expressing tumours, with $7.22 \times 10^4$ or $7.16 \times 10^4$ antigen-binding capacity (ABC) per cell, respectively, and MKN74 and MKN45 cells to represent low GPC3-expressing tumours, with $2.49 \times 10^3$ or $4.86 \times 10^3$ ABC per cell, respectively (Fig. 1a). As the CD3 arm of ERY974 does not cross-react with murine CD3, we employed a humanised mouse model (huNOG) where human T cells are differentiated and constitutively supplied through administrated human CD34 + stem cells[22] to evaluate the antitumour activity of ERY974. As shown in Fig. 1b, ERY974 induced tumour regression in PC10 and MKN74 xenografts (105 and 132% of tumour growth inhibition (TGI) activity, respectively), but not in NCI-H446 and MKN45 xenografts (50 and 37% of TGI activity, respectively), suggesting that factors apart from GPC3 expression are important for determining ERY974 potency. To evaluate the alloreactive effect of T cells differentiated

from allogeneic CD34 + stem cells on tumour growth, we compared the antitumour efficacy of ERY974 against PC10 and NCI-H446 in huNOG mice administered CD34 + stem cells from different donors. The TGI of ERY974 in PC10 xenografts was 104% and 105% (Supplementary Fig. 1a) and in NCI-H446 xenografts was 38% and 49% (Supplementary Fig. 1b), respectively. Furthermore, we confirmed that the tumour growth of NCI-H446 xenografts with or without administration of allogeneic ex vivo-expanded T cells is comparable (Supplementary Fig. 1c). These data suggest that the effect of alloreactive T cells on tumour growth could be negligible. From our data showing different TGI despite comparable GPC3 expression, we speculated that the presence of tumour-infiltrating immune cells at baseline may be critical to improve sensitivity of tumour to ERY974. To test this hypothesis, we determined human and mouse CD45 scores at baseline using RNA sequence data and found that this score is closely correlated with the efficacy of ERY974 (Fig. 1c). We also conducted immunohistochemistry (IHC) analysis for human CD3 in these tumours in vehicle- or ERY974-treated mice. CD3-positive cells were more abundant in PC10 but not in other tumours at baseline and further increased in PC10 and MKN74 after ERY974 administration; however, there was only a slight increase in the number of CD3-positive cells in NCI-H446 and MKN45 tumours after ERY974 administration (Supplementary Fig. 2a-c). Interestingly, in MKN74, CD3-positive cells were not abundant at baseline, but they increased significantly following ERY974 administration. We observed that the expression of human CD14, a myeloid cell marker, was significantly higher in MKN74 tumours at baseline, indicating that immune cells at baseline may not be T cells, but myeloid cells (Supplementary Fig. 2d). We speculated that residential myeloid cells may help to increase the number of T cells after ERY974 treatment by secreting cytokines or chemokines. Consistent with this, the expression of T cell chemokine genes such as *CCL3*, *CCL4*, *CCL5*, *CXCL9*, *CXCL10* and *CXCL11*[23] in MKN74 tumours tends to be higher than that in other tumours, especially after ERY974 administration (Supplementary Fig. 2e). These results demonstrate that the degree of infiltration by immune cells at baseline is one of the factors in determining the in vivo efficacy of ERY974.

### ERY974+ chemotherapy increases antitumour efficacy in non-inflamed tumours

To increase the efficacy of ERY974 against NCI-H446 or MKN45 tumours with low immune cell infiltration (i.e. non-inflamed tumour model), we tested combination therapy of ERY974 with chemotherapy drugs used in the treatment of lung cancer for NCI-H446 (lung cancer) tumours, and chemotherapy drugs used in the treatment of gastric cancer for MKN45 (gastric cancer) tumours. Both, lung cancer and gastric cancer are considered to be ERY974 target tumour types. We selected cisplatin, paclitaxel and capecitabine as representative chemotherapy drugs because cisplatin and paclitaxel are used as the standard of care in the first-line treatment of lung cancer[24,25], and fluorouracil (5'FU)-related drugs, including capecitabine are used as the standard of care in the first-line treatment of gastric cancer[26]. These drugs were tested in combination with ERY974 in the huNOG model as well as a T cell-injected model, in which ex vivo-expanded human T cells are injected as effector cells in immunodeficient mice. ERY974 monotherapy showed modest antitumour effects, while paclitaxel monotherapy showed strong efficacy at first but was unable to prevent eventual tumour regrowth. Meanwhile, the combination of ERY974 with paclitaxel achieved a complete response in NCI-H446 lung tumour in both huNOG and T cell-injected models (Fig. 2a, b), and similar results were obtained in MKN45 gastric tumour in T cell-injected model (Supplementary Fig. 3a). Improved antitumour efficacy was also observed when ERY974 was combined with cisplatin in NCI-H446 (Fig. 2c, d) or MKN45 (Supplementary Fig. 3b) tumours. Efficacy was also improved when ERY974 was

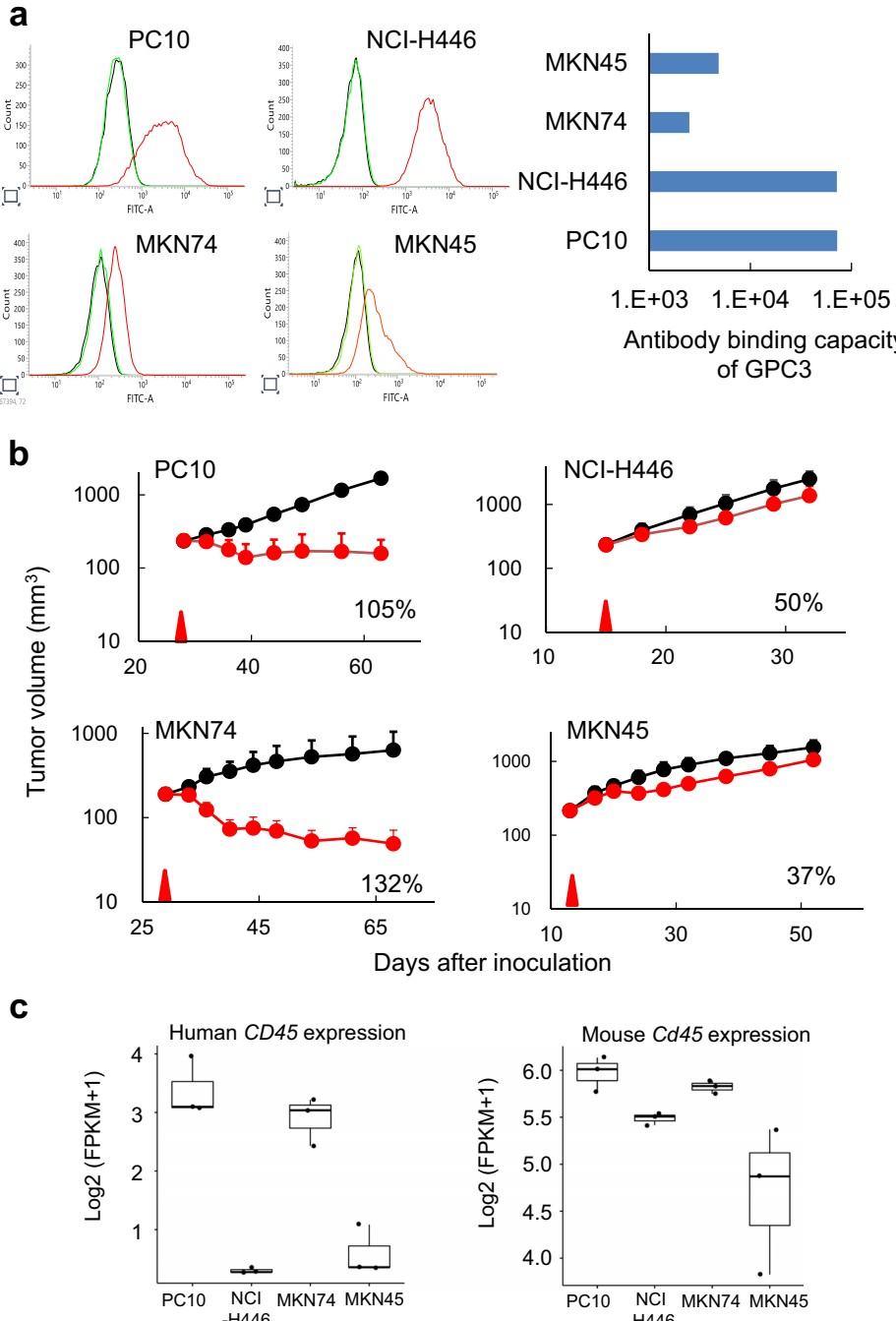

**Fig. 1 | The number of infiltrated immune cells at baseline is a predictive marker of ERY974 efficacy in the huNOG model. a** (Histogram) Cell surface expression of GPC3 determined using flow cytometry with anti-GPC3 antibody ($n = 1$). Black line indicates a shift with no antibody; the green line indicates a shift with isotype control; the red line indicates a shift with an anti-GPC3 antibody. (Table) Antibody binding capacity (ABC) of GPC3 in various cell lines with GPC3-high (PC10 and NCI-H446) and GPC3-low (MKN74 and MKN45) expression. **b** Antitumour efficacy of ERY974 (1 mg/kg) in various tumour models ($n = 5$). During studies, one mouse in the vehicle group in the PC10 model was sacrificed due to body weight loss on day 44, and one mouse in the ERY974 group in the MKN74 model was sacrificed due to body weight loss on 36. The red arrow indicates the timing of ERY974 administration. TGI value is shown in each figure. Data were shown as the mean ± standard deviation (SD). **c** Human and mouse *CD45* FPKM score at baseline is a predictive marker of ERY974 efficacy. RNAseq was conducted on total RNA extracted from PC10, NCI-H446, MKN74 and MKN45 tumour xenografts of huNOG mice ($n = 3$). The FPKM + 1 scores of human and mouse *CD45* were shown as log2-transformed value. In the boxplot, centre lines show median values, box limits show upper and lower quartiles, and whiskers show minimum and maximum values. Source data are provided as a Source Data file.

combined with capecitabine in MKN45 (Fig. 2e, f) or NCI-H446 tumours, in which a statistically significant difference was not observed between ERY974 and combination groups, however, the combination group showed higher TGI on day 32 than the mono-therapy group (TGI: combination; 94%, ERY974; 80% and capecita-bine; 11%) (Supplementary Fig. 3c).

## ERY974+ paclitaxel promotes T cell infiltration in non-inflamed NCI-H446 tumours

To investigate the mechanism underlying the synergistic antitumour effect of ERY974 in combination with paclitaxel, we administered 20 mg/kg paclitaxel on day −1 and 5 mg/kg ERY974 on day 0 in huNOG mice inoculated with NCI-H446 cells, followed by tumour

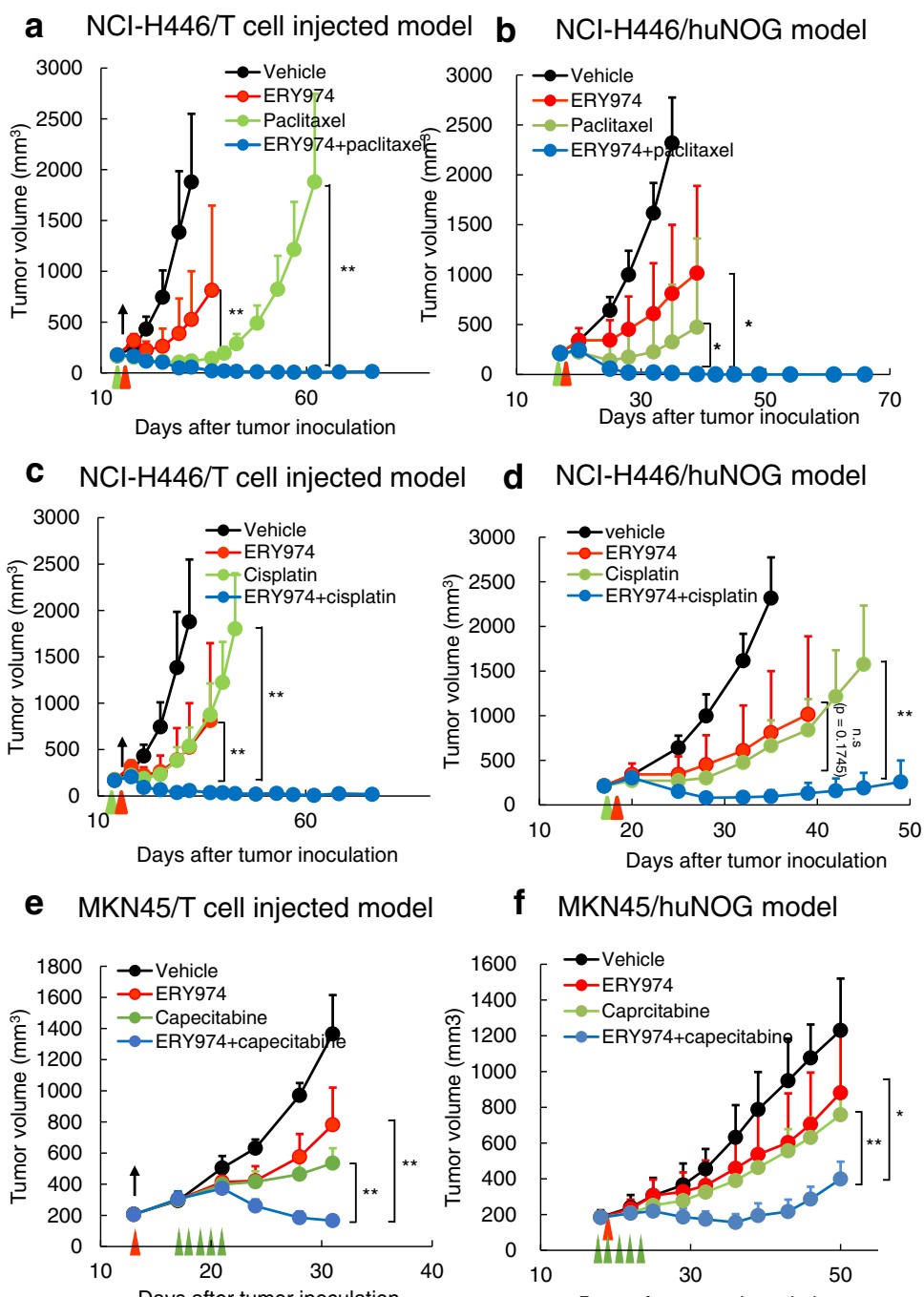

**Fig. 2 | Combination of ERY974 and chemotherapy shows more efficacy than ERY974 or chemotherapy alone in both huNOG and T cell-injected models.**
**a**, **b** ERY974 (1 and 5 mg/kg in **a** and **b**, respectively) + paclitaxel (20 mg/kg) efficacy in NCI-H446 xenograft tumours in the T cell-injected (**a**) and huNOG (**b**) models (*n* = 5). During study, one mouse in the ERY974 + paclitaxel group in (**b**) was sacrificed due to body weight loss on day 35. **c**, **d** ERY974 (1 and 5 mg/kg in **c** and **d**, respectively) + cisplatin (7.5 mg/kg) efficacy in NCI-H446 xenograft tumours in the T cell-injected (**c**) and huNOG (**d**) models (*n* = 5). During study, one mouse in the ERY974 + cisplatin group in (**c**) was sacrificed due to body weight loss on day 53. **e**, **f** ERY974 (5 mg/kg) + capecitabine (431 and 359 mg/kg in **e** and **f**, respectively) efficacy in MKN45 xenograft tumours in the T cell-injected (**e**) and huNOG (**f**) models. Red and green arrows indicate the timing of ERY974 and chemotherapy

drug administration, respectively. Black arrows indicate the timing of T cell administration. Tumour volumes are presented as the mean ± SD (*n* = 5). *$P < 0.05$, **$P < 0.01$, ***$P < 0.001$, n.s. no significance (Wilcoxon chi-square test). Exact *p* values are as follows. In Fig. 2a, *p* value of ERY974 versus combination on day 37 is 0.0082, and *p* value of paclitaxel versus combination on day 62 is 0.0071. In Fig. 2b, *p* value of ERY974 versus combination and *p* value of paclitaxel versus combination on day 39 are both 0.0127. In Fig. 2c, *p* value of ERY974 versus combination on day 37 is 0.0082 and *p* value of cisplatin versus combination on day 43 is 0.0088. In Fig. 2d, *p* value of ERY974 versus combination on day 39 is 0.1745 and *p* value of cisplatin versus combination on day 45 is 0.0088. In Fig. 2e, *p* value of ERY974 versus combination and *p* value of capecitabine versus combination, on day 50 are 0.0163 and 0.0090, respectively. Source data are provided as a Source Data file.

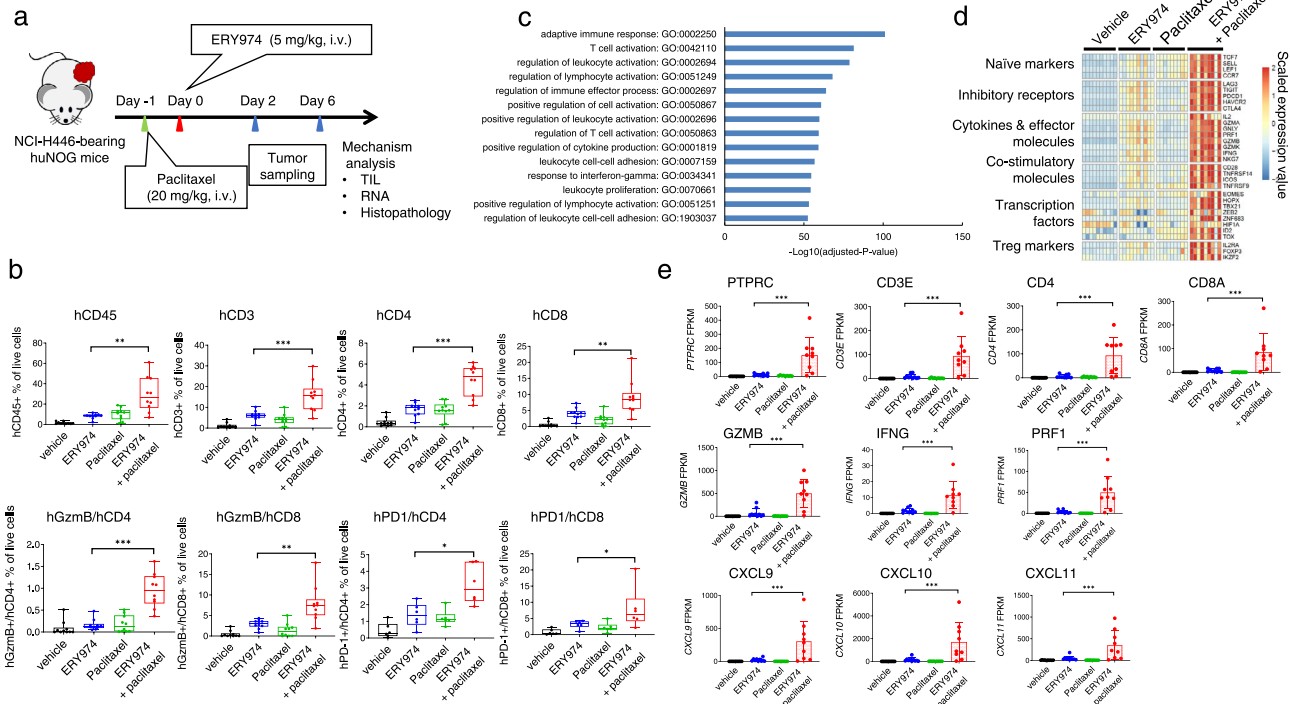

**Fig. 3 | Infiltration and activation of T cells are enhanced in the ERY974 + paclitaxel combination in non-inflamed NCI-H446 tumours. a** Schematic illustrating the experimental setup. For TIL and RNA analysis of day 6 samples, the experiment was conducted twice, and the data were combined. **b** Analysis of tumour-infiltrating lymphocytes (TILs) isolated on day 6 from NCI-H446 tumours using flow cytometry ($n = 10$ except for data of hPD-1 + hCD4+ and hPD-1 + hCD8 + : $n = 6$). In the boxplot, centre lines, box limits, and whiskers show median values, upper and lower quartiles, and minimum and maximum values, respectively. Statistical significance was determined using two-tailed unpaired $t$-tests (*$P < 0.05$, **$P < 0.01$, ***$P < 0.001$, n.s. no significance). Exact $p$ values of ERY974 versus combination in % of live cells for hCD45+, hCD3+, hCD4+, hCD8+, hGzmb+/hCD4+, hGzmb+/hCD8+, hPD-1+/CD4+ and hPD-1+/hCD8+ are 0.0056, 0.0006, 0.0001, 0.0043, 0.0001, 0.0029, 0.0104 and 0.0374, respectively. **c** Gene ontology terms associated with genes upregulated in ERY974-treated groups. Differential expression analysis between vehicle and ERY974-treated groups was performed with

RNAseq data from day 6 samples ($n = 10$). Gene set analysis of differentially expressed genes was performed using Fisher's exact test. False discovery rate was calculated using the Benjamini–Hochberg method for multiple testing corrections. **d** Heatmap of T cell signatures of vehicle, paclitaxel, ERY974, and paclitaxel + ERY974 groups. RNAseq data of day 6 samples ($n = 9$ for paclitaxel, combination groups, or $n = 10$ for vehicle and ERY974 groups) were used. The heatmap was created by calculating the $z$-scores using log2-transformed FPKM values for all target genes. **e** mRNA levels of representative genes for T cell marker, T cell activation marker, cytokine and chemokine. RNAseq data of day 6 samples ($n = 9$ or 10, described above) were used. Data were presented as the mean ± SD. Statistical significance was determined using edgeR glmQLFTest (*$P < 0.05$, **$P < 0.01$, ***$P < 0.001$). Exact $p$ values of ERY974 versus combination for *PTPRC*, *CD3E*, *CD4*, *CD8A*, *GZMB*, *IFNG*, *PRF1*, *CXCL9*, *CXCL10* and *CXCL11* are $2.62 \times 10^{-7}$, $1.00 \times 10^{-5}$, $2.86 \times 10^{-7}$, $2.62 \times 10^{-6}$, $4.63 \times 10^{-5}$, $9.88 \times 10^{-5}$, $5.01 \times 10^{-7}$, $1.51 \times 10^{-6}$, $7.49 \times 10^{-7}$ and $1.91 \times 10^{-6}$, respectively. Source data are provided as a Source Data file.

collection on days 2 and 6 for tumour-infiltrating lymphocyte (TIL), RNA and histopathological analyses (Fig. 3a). On day 2, the number of hCD3-positive cells was slightly increased in the ERY974 group but was not further increased in the combination group according to flow cytometry analysis (Supplementary Fig. 4a). Representative gating strategy of flow cytometry analysis is shown in Supplementary Fig. 4b. On day 6, the number of hCD3- hCD4 or hCD8-positive cells was higher in the ERY974 group than that on day 2, and was significantly increased in the combination group compared with the ERY974 monotherapy group. Moreover, the numbers of double-positive cells for T cell markers and T cell activation markers, such as Granzyme B (GzmB) and PD-1, were higher in the combination group than in the ERY974 group. These data suggest that combination therapy promotes infiltration and activation of T cells in the tumour by day 6 (Fig. 3b). Next, we conducted gene set analyses based on the differentially expressed genes using Fisher's exact test to identify genes affected by ERY974 monotherapy. Enriched genes that were upregulated after ERY974 treatment were related to T cell activation and response (Fig. 3c). We thus selected T cell-related genes for generating a heatmap using our RNA data from day 6. The expression of ERY974-induced T cell activation-related genes in the ERY974 monotherapy group was further increased in the combination group

(Fig. 3d). Moreover, the RNA expressions of representative genes for T cell markers, T cell activation markers, cytokines, and chemokines were significantly higher in the combination group than that in the monotherapy group (Fig. 3e).

## ERY974 + cisplatin promotes T cell infiltration in non-inflamed NCI-H446 tumours

We also analyzed the mechanism underlying the combined antitumour effect of ERY974 and cisplatin against NCI-H446 tumours (Supplementary Fig. 5a). TIL analysis demonstrated that ERY974 + cisplatin significantly increased the number of hCD3-, hCD4-, hCD8 and hCD45-positive cells in the tumour, and the number of GzmB/CD4, GzmB/CD8, PD-1/CD4 and PD-1/CD8-double-positive cells was higher in the combination group than that in the ERY974 monotherapy group (Supplementary Fig. 5b). Heatmap analysis showed that there was a greater increase in the expression of T cell marker, and T cell activation-related genes in the combination group than that in the ERY974 monotherapy group (Supplementary Fig. 5c). Moreover, the expression of genes for T cell markers, T cell activation markers, and T cell chemokines was also higher in the combination group than that in the monotherapy group (Supplementary Fig. 5d).

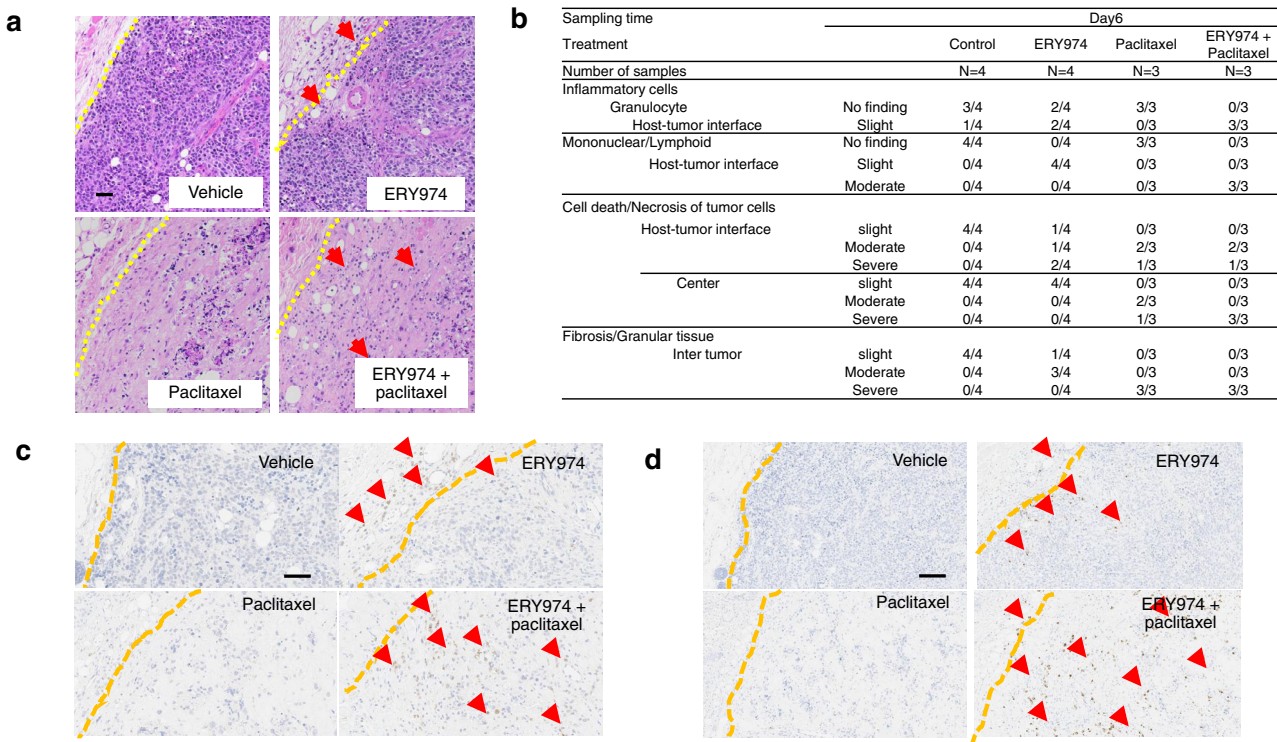

| Sampling time | | | | Day6 | |
|---|---|---|---|---|---|
| Treatment | | Control | ERY974 | Paclitaxel | ERY974 + Paclitaxel |
| Number of samples | | N=4 | N=4 | N=3 | N=3 |
| **Inflammatory cells** | | | | | |
| Granulocyte | No finding | 3/4 | 2/4 | 3/3 | 0/3 |
| Host-tumor interface | Slight | 1/4 | 2/4 | 0/3 | 3/3 |
| Mononuclear/Lymphoid | No finding | 4/4 | 0/4 | 3/3 | 0/3 |
| Host-tumor interface | Slight | 0/4 | 4/4 | 0/3 | 0/3 |
| | Moderate | 0/4 | 0/4 | 0/3 | 3/3 |
| **Cell death/Necrosis of tumor cells** | | | | | |
| Host-tumor interface | slight | 4/4 | 1/4 | 0/3 | 0/3 |
| | Moderate | 0/4 | 1/4 | 2/3 | 2/3 |
| | Severe | 0/4 | 2/4 | 1/3 | 1/3 |
| Center | slight | 4/4 | 4/4 | 0/3 | 0/3 |
| | Moderate | 0/4 | 0/4 | 2/3 | 0/3 |
| | Severe | 0/4 | 0/4 | 1/3 | 3/3 |
| **Fibrosis/Granular tissue** | | | | | |
| Inter tumor | slight | 4/4 | 1/4 | 0/3 | 0/3 |
| | Moderate | 0/4 | 3/4 | 0/3 | 0/3 |
| | Severe | 0/4 | 0/4 | 3/3 | 3/3 |

**Fig. 4 | ERY974 combined with paclitaxel increases infiltration of T cells and macrophages into the centre of the non-inflamed NCI-H446 tumour.** Tumour tissues were prepared from NCI-H446 xenograft tumours of huNOG mice ($n = 3$ for paclitaxel and ERY974 + paclitaxel groups; $n = 4$ for vehicle and ERY974 groups). **a** HE staining. Yellow dotted line indicates tumour-stromal boundary; arrows indicate inflammatory cells. Scale bar is 100 μm **b** Histopathological analysis with HE staining. **c**, **d** Representative IHC for hCD3 (**c**) and hCD68 (**d**). Yellow dotted line indicates tumour-stromal boundary; arrows indicate hCD3-positive (**c**) and hCD68-positive (**d**) cells. Scale bar is 100 μm (**c**) and 200 μm (**d**).

## ERY974 + paclitaxel enhances immune cell infiltration into the centre of non-inflamed NCI-H446 tumour

Next, we conducted a histopathological analysis of day 6 samples to examine the mechanism of synergy between ERY974 and paclitaxel. Haematoxylin and eosin (HE) staining data showed that, unlike in the vehicle and paclitaxel groups, there was an increase in the number of inflammatory cells, including granulocytes and lymphocytes, in the tumours in the ERY974 monotherapy group, but infiltration of inflammatory cells was limited to the tumour-stromal boundary. However, in the ERY974 and paclitaxel combination group, inflammatory cells infiltrated inside the tumour and increased in number therein (Fig. 4a). Consistent with this, combination therapy caused severe cell death or necrosis of tumour cells in the centre of the tumour (Fig. 4b). We then performed IHC for hCD3 (human T cell marker) and hCD68 (human macrophage marker) to identify the type of inflammatory cells infiltrating the tumour. In the ERY974 group, a small number of CD3-positive cells were present only at the tumour-stromal boundary. However, the ERY974 + paclitaxel group showed an increased number of CD3-positive cells that also infiltrated inside the tumour (Fig. 4c); similar data were obtained for hCD68 (Fig. 4d). These data indicate that the non-inflamed NCI-H446 tumour structure was too desmoplastic for inflammatory cells to easily infiltrate into the tumour centre (Fig. 4a); however, paclitaxel disrupted the tumour structure, allowing inflammatory cells to reach the tumour centre.

## ERY974 delivery in non-inflamed NCI-H446 tumours is improved by paclitaxel

Based on the immune cell distribution data in Fig. 4, we hypothesised that the combination of ERY974 and paclitaxel also improves the distribution of ERY974 in tumours. To test this, we conducted a distribution study using FITC-labelled ERY974. First, we confirmed that FITC-labelled ERY974 is comparable to unlabelled ERY974 in terms of binding activity to both GPC3 and CD3 antigens (Supplementary Fig. 6a) and cytotoxic activity in SK-pca13a that was engineered to express GPC3 in SK-HEP-1, a GPC3-negative cell line (Fig. 5a). FITC-labelled ERY974 was then injected into huNOG mice inoculated with NCI-H446 cells 1 day after paclitaxel administration (Fig. 5b). Tumour and plasma samples were collected at 24, 72 and 144 h after antibody administration, and the fluorescence intensity of FITC was measured. The plasma concentration of FITC-labelled ERY974 did not differ between groups of FITC-labelled ERY974 alone and its combination with paclitaxel (Fig. 5c). However, the distribution of FITC-labelled ERY974 in the tumour significantly increased at 72 and 144 h post-dose in the combination group compared with FITC-labelled ERY974 alone group (Fig. 5d). These findings indicate that combination therapy improves the distribution of ERY974 in addition to immune cells in the tumour, explaining the synergistic effect of ERY974 + paclitaxel in non-inflamed NCI-H446 tumours (Supplementary Fig. 7).

## ERY974 + capecitabine promotes T cell infiltration in non-inflamed MKN45

We next examined the mechanism underlying the synergistic anti-tumour effects of ERY974 + capecitabine in non-inflamed MKN45 tumours. Capecitabine was orally administered for 4 or 5 consecutive days starting from day −1 and ERY974 was intravenously administered on day 0 in huNOG mice inoculated with MKN45 cells (Fig. 6a). Tumours were collected on days 3, 7 and 14 for RNA analysis using nCounter. The synergistic pharmacodynamic (PD) effect was the most evident in day 14 samples, in which only the combination group showed statistical significance against the vehicle group in expression of genes of human tyrosine phosphatase receptor type C (*PTPRC*) and h*CD3* (Fig. 6b). This suggests that the optimum time for detecting a combination effect differs depending on the chemotherapy drugs, although an increase in immune cells is likely to be

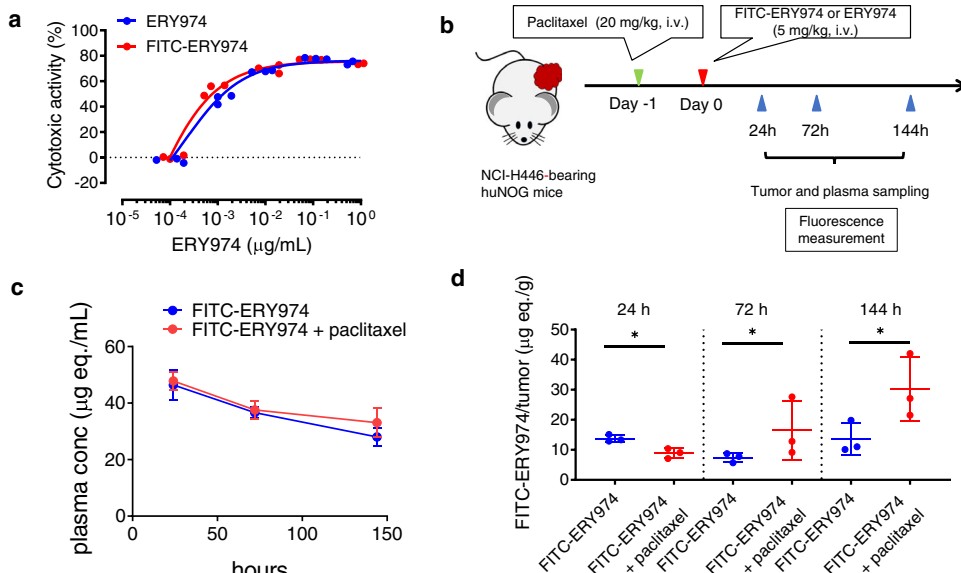

**Fig. 5 | Distribution of FITC-labelled ERY974 in non-inflamed NCI-H446 tumours is improved in combination therapy with paclitaxel. a** TDCC analysis of non-labelled and FITC-labelled ERY974 in SK-pca13a cell (*n* = 3). **b** Schematic summarising the distribution study. **c**, **d** Concentration of non-labelled and FITC-labelled ERY974 in plasma (**c**) and tumours (**d**). Data were expressed as the mean ± SD (*n* = 3). Statistical analysis was conducted using Wilcoxon rank-sum chi-square test (*$P < 0.05$). Exact *p* values of FITC-ERY974 versus FITC-ERY974 + paclitaxel for 24, 72 and 144 h are all 0.0495. Source data are provided as a Source Data file.

the common mechanism underlying the additive or synergistic effects observed with the three chemotherapies tested here. To comprehensively understand the alterations in gene expression, we conducted RNAseq on day 14 samples. Heatmap analysis revealed an apparent combination effect (Fig. 6c). Moreover, gene expression for T cell marker, T cell activation, and chemokine was significantly higher in the combination group than that in the ERY974 monotherapy group (Fig. 6d).

### ERY974 converts capecitabine to its active form by inducing thymidine phosphorylase expression

So far, we have demonstrated that chemotherapy helps to improve ERY974-induced antitumour efficacy by altering the TME into a favourable environment for the distribution of both immune cell and ERY974. Next, we examined whether ERY974 has a reciprocal effect on the chemotherapy-induced antitumour efficacy. Chemotherapy alters the expression of genes associated with mitosis; thus, we compared the enrichment scores of the Gene Ontology (GO) terms for apoptosis, cell cycle, and DNA replication between the chemotherapy alone and combination groups. The enrichment score of the ERY974 + chemotherapy group was higher in GO for apoptosis and lower for cell cycle and DNA replication than that of the chemotherapy alone group, suggesting that ERY974 may have an influence on chemotherapy-induced mitosis inhibition, although the impact of ERY974 on the enrichment score is more apparent in cisplatin or capecitabine combination than that in paclitaxel combination (Fig. 7a and Supplementary Fig. 8a, b). These results indicate that ERY974 enhances capecitabine- or cisplatin-induced mitosis inhibition, resulting in synergistic efficacy. To further elucidate this mechanism, we focused on the ERY974 + capecitabine combination. Capecitabine is a pro-drug that is absorbed from the intestine and metabolised into 5′-deoxy-5-fluorocytidine (5′-DFCR) in the liver by carboxylesterase, then further metabolised to 5′-deoxy-5-fluorouridine (5′-DFUR) in the liver or tumours by cytidine deaminase, and finally converted to fluorouracil (5′-FU)−its active form−by thymidine phosphorylase (TP), which is highly expressed in tumours[27] (Supplementary Fig. 9a). We found that TP expression is increased in both the ERY974 alone and combination groups, as evidenced by

western blotting. Interestingly, high TP expression levels were sustained only in the combination group of day 14 samples (Fig. 7b). We also found that an expression of *TYMP* gene which encodes TP is higher in the combination group (Fig. 7c). These data suggest that sustained TP expression in the combination group promotes capecitabine conversion to 5′-FU in tumours, thereby explaining the mechanism of combination effect.

Various cytokines, such as interferon-gamma (IFNγ) and tumour necrosis factor-alpha (TNFα), are known to induce TP expression[28]. To further examine whether ERY974-induced cytokine expression could increase TP expression in vitro, we performed a T cell-dependent cellular cytotoxicity (TDCC) assay in MKN45 cells (Fig. 7d), followed by western blotting. As expected, TP expression in MKN45 cells was increased after the TDCC assay (Fig. 7e), where ERY974 activated T cells to secrete IFNγ, TNFα (Fig. 7f), and other cytokines (Supplementary Fig. 9b). Accordingly, western blotting (Fig. 7g) and qRT-PCR analysis (Fig. 7h) revealed that recombinant IFNγ or TNFα upregulated expression of both TP protein and *TYMP* gene in MKN45 cells. This is not specific to MKN45, as the induction of both TP protein and *TYMP* gene expression by recombinant IFNγ or TNFα is observed in NCI-H446, PC10, and MKN74 cells, although baseline protein levels vary among the cell lines (Supplementary Fig. 9c, d). MKN45 was not highly sensitive to 5′-DFUR, the 5′-FU precursor and direct substrate of TP (Supplementary Fig. 9a), as TP expression is low in these cells. However, IFNγ or TNFα sensitised MKN45 to 5′-DFUR, indicating that 5′-DFUR is converted to 5′-FU via the IFNγ- or TNFα−induced TP enzyme (Fig. 7i, j). Analysis of the RNA data using nCounter showed a greater increase in the signature scores of the genes involved in the cell-cycle arrest and the DNA damage repair in the 5′-DFUR and IFNγ combination group compared with the 5′-DFUR or IFNγ alone group (Supplementary Fig. 9e), while the signature score of the mitogen-activated protein kinase was decreased in the combination group. Overall, these data show that IFNγ and TNFα secreted from T cells during TDCC can induce *TYMP* gene transcription followed by TP protein synthesis in MKN45, which efficiently converts capecitabine to 5′-FU at the tumour site. These findings explain the mechanism of how ERY974 enhances capecitabine-induced cytotoxic activity.

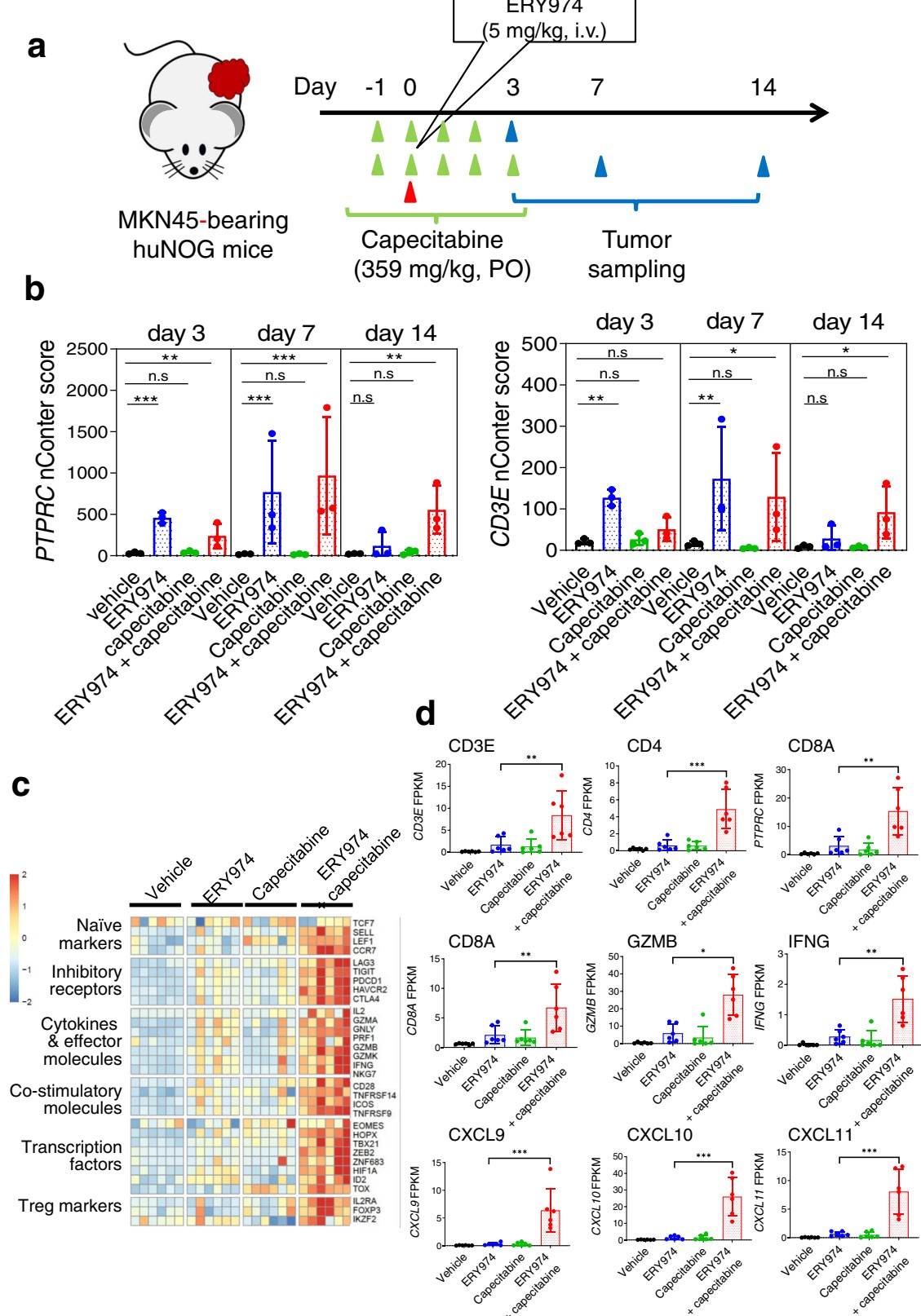

**EGFR-TRAB is utilised to generalise the combination effect of TRAB and chemotherapy**

To examine if our findings could be generalised for TRABs other than ERY974, we attempted to prepare for a TRAB targeting CD3 and different antigen from GPC3 and conduct studies on their efficacy in combination with chemotherapy drugs. We selected EGFR as the representative tumour antigen for TRAB because EGFR is one of the most well-known oncogenes[29] and therefore suitable to demonstrate whether our findings are generalisable. We designed EGFR-TRAB with CrossMab technology[30] (Fig. 8a) using the complementarity-determining region (CDR) of cetuximab[31]. First, we conducted a surface plasmon resonance (SPR) analysis to study the binding kinetics. As

**Fig. 6 | Infiltration and activation of T cells are enhanced in the ERY974 + capecitabine combination in non-inflamed MKN45 tumours. a** Schematic illustrating the experimental setup for RNA analysis using nCounter ($n = 3$). Only day 14 experiments were repeated with $n = 3$ per group, and the combined RNAseq data is presented in (**c**) and (**d**). **b** *PTPRC* and *CD3E* nCounter score in the indicated groups and at different time points ($n = 3$). Data were presented as the mean ± SD. Statistical significance was evaluated using the two-sided Dunnet test. Exact $p$ values of vehicle versus ERY974, vehicle versus capecitabine, and vehicle versus combination for *PTPRC* on day 3 are <0.001, 0.736 and 0.0013, respectively. Similarly, exact $p$ values of *PTPRC* on day 7 are <0.001, 0.957 and <0.001, respectively. Similarly, exact $p$ values of *PTPRC* on day 14 are 0.565, 0.921 and 0.00907, respectively. Exact $p$ values of vehicle versus ERY974, vehicle versus capecitabine, and vehicle versus combination for *CD3E* on day 3 are 0.00132, 0.920 and 0.0887, respectively.

Similarly, exact $p$ values of *CD3E* on day 7 are 0.00375, 0.0954 and 0.0111, respectively. Similarly, exact $p$ values of *CD3E* on day 14 are 0.341, 0.997 and 0.0107, respectively. **c** Heatmap of T cell signatures using RNAseq data of day 14 samples. Z-scores were calculated using log2-transformed FPKM values for all target genes. **d** mRNA levels of representative genes for T cell marker, T cell activation marker, cytokine and chemokine using RNAseq data of day 14 samples ($n = 6$). The FPKM scores were compared between each group. Data were presented as the mean ± SD. Statistical significance was evaluated using edgeR glmQLSTest (*$P < 0.05$, **$P < 0.01$, ***$P < 0.001$, and n.s. no significance). Exact $p$ values of ERY974 versus combination for *CD3E*, *CD4*, *CD8A*, *PTPRC*, *GZMB*, *IFNG*, *CXCL9*, *CXCL10* and *CXCL11* are $8.27 \times 10^{-3}$, $2.65 \times 10^{-4}$, $1.66 \times 10^{-3}$, $5.01 \times 10^{-3}$, $2.20 \times 10^{-2}$, $6.21 \times 10^{-3}$, $6.32 \times 10^{-7}$, $3.82 \times 10^{-7}$ and $1.44 \times 10^{-5}$, respectively. Source data are provided as a Source Data file.

shown in Fig. 8b and Supplementary Fig. 10a and b, the $K_D$ values of EGFR-TRAB and cetuximab to EGFR are similar ($4.58 \times 10^{-9}$ M for EGFR-TRAB, $4.98 \times 10^{-9}$ M for cetuximab), as are the $K_D$ values of EGFR-TRAB and ERY974 to CD3 ($8.59 \times 10^{-7}$ M for EGFR-TRAB, and $2.07 \times 10^{-7}$ M for ERY974[15]) (Supplementary Fig. 10c), proving that we were able to successfully engineer the EGFR-TRAB. EGFR is expressed in MKN45 (ABC: $3.48 \times 10^{4}$), but not in NCI-H446 (Fig. 8c). Therefore, we focused further analysis on MKN45 (Fig. 8b), where EGFR-TRAB showed substantial TDCC (Fig. 8d).

### EGFR-TRAB + chemotherapy shows synergistic efficacy in non-inflamed MKN45 tumour

Then we examined in vivo efficacy of EGFR-TRAB in MKN45 tumour in T cell-injected model. In contrast to in vitro activity, the antitumour activity of EGFR-TRAB monotherapy in MKN45 was moderate due to the non-inflamed TME of MKN45 (Fig. 9a, b). Nevertheless, EGFR-TRAB combined with paclitaxel or capecitabine showed a statistically significant increase in efficacy (Fig. 9a, b). Moreover, we conducted a PD analysis of the EGFR-TRAB + paclitaxel combination. We administered paclitaxel on day −1, and EGFR-TRAB on day 0, then the MKN45 tumour was collected on day 6 for RNA analysis (Fig. 9c). We confirmed that the expression of representative genes of the T cell marker, T cell activation, and chemokine was significantly higher in the combination group than that in the monotherapy groups (Fig. 9d). These data indicate that the enhanced efficacy of TRAB combined with chemotherapy may not be limited to only ERY974, but could be a broader phenomenon, also observable in TRABs targeting other cancer antigens.

### Discussion

ERY974 induces tumour regression in a variety of tumour models[15], but shows only moderate efficacy in non-inflamed tumours with a low number of immune cells at baseline, such as NCI-H446 and MKN45. Herein, we found that a combination of ERY974 and chemotherapy significantly improved efficacy in non-inflamed tumours by increasing the number of T cells in the tumour. We also found that T cells remained at the tumour-stromal boundary with ERY974 treatment alone but were able to infiltrate to the centre of the tumour mass when combined with paclitaxel, which also increased the distribution of ERY974 at the tumour site. In turn, ERY974 increased the chemotherapy-induced up- or downregulation of expression for mitosis-related genes. As one of the mechanisms explaining this, ERY974 was found to promote capecitabine conversion into its active 5′-FU form by inducing TP enzyme expression in tumours. Overall, our findings demonstrate that ERY974 and chemotherapy reciprocally promote antitumour efficacy in non-inflamed tumours, representing a promising strategy for patients with non-inflamed tumours.

Notably, ERY974 monotherapy was, to some extent, effective against non-inflamed tumours, such as NCI-H446 and MKN45 (Fig. 1b). These results are consistent with a previous study that used a syngeneic model to demonstrate that ERY974 shows significant

antitumour efficacy even in LLC1/hGPC3 tumours, which are representative of non-inflamed tumours[32] and where anti-PD-1, PD-L1, and CTLA4 antibodies failed to show efficacy[15]. We speculate that this is because TRAB functions independently of MHC class I and TCR complex formation. When the number of cytotoxic lymphocytes recognising tumour-derived peptides presented by MHC class I is limited, the TME likely becomes non-inflamed, hindering the efficacy of ICIs. However, ERY974 can utilise any kind of T cell as an effector cell only if CD3 is expressed, thereby inducing the proliferation of a variety of T cells at the tumour site[15]. Furthermore, unlike ICI, ERY974 can exert TDCC (Figs. 5a, 7d) and antitumour efficacy (Fig. 2a–f), utilising allogeneic effector cells. These observations strongly suggest that TRAB modality can satisfy unmet medical needs by showing efficacy in non-inflamed tumours.

In our gene expression analysis, ERY974 upregulated a variety of genes whose expression level differed among subsets of T cells, including naïve T cells, cytotoxic T cells, and Tregs (Figs. 3d, 6c and Supplementary Fig. 4c). These data suggest that ERY974 could simultaneously induce activation and proliferation of various subsets of T cells. Consistent with this, it was recently reported that blinatumomab, a BiTE antibody targeting CD3 and CD19, activated CD8 + effector memory T cells, CD4 + central memory T cells, naïve T cells and Tregs at the same time in single-cell RNA analysis in vitro[33]. Similarly, tebentafusp can utilise various T cells, including CD45RA effector memory, central memory, effector memory, and naïve T cells, as effector cells[34]. We believe that TRABs might not have a preference for subtypes of T cells only if they express CD3. Alternatively, ERY974 may indirectly induce tumour infiltration and activation of various immune cells through cytokines or chemokines secreted from T cells that are initially activated by ERY974.

Based on the description above, we believe that ERY974 monotherapy may show efficacy in tumours with differing TMEs. Nonetheless, in predicting the efficacy of ERY974, we must consider the number of immune cells infiltrating the tumour at baseline, since ERY974 achieved tumour regression in inflamed tumours (PC10 and MKN74) but not in non-inflamed tumours (NCI-H446 and MKN45), despite displaying moderate antitumour activity (Fig. 1a). To achieve a complete response against such non-inflamed tumours, a different strategy is needed. We showed that in combination therapy, paclitaxel disrupts the rigid tumour structure, making the TME favourable for ERY974 and allowing T cells to infiltrate the tumour more efficiently and increasing ERY974 distribution at the tumour site (Supplementary Fig. 7). These events may similarly occur in the combination of chemotherapy drugs with most TRABs, as we obtained similar data when we used EGFR-TRAB (Fig. 9). According to our protocol of this study, we usually completed tumour size measurement of the group where tumour size in one of the mice reaches approximately 2000 mm³, which caused variability of observation periods among groups, and an unavailability to draw survival curves. This may cause a limitation of profound assessment for antitumour effect of the combination. However, we could show that TGI is apparently higher in combination

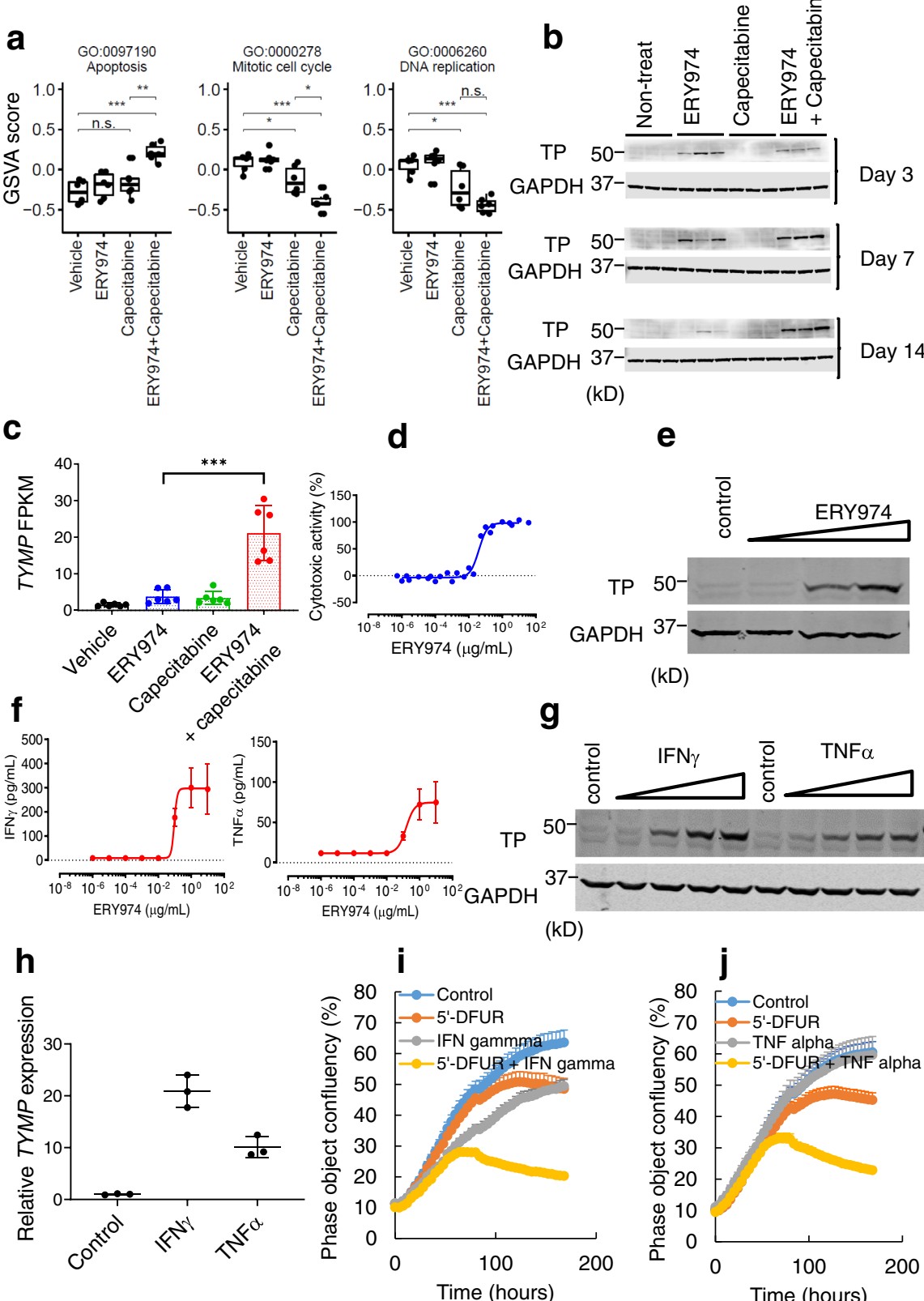

group than that of monotherapy groups in all the mon-inflamed tumour models.

Chemotherapy drugs are supposed to show cytotoxicity in immune cells as well as in tumour cells. However, we observed increased antitumour efficacy in the combinations of chemotherapy drugs with ERY974, for which T cells are essential to exert cytotoxicity.

We speculate that the ability of ERY974 to induce proliferation of T cells could eventually overcome the initial inhibitory effect of chemotherapy. ERY974 may also be recruiting freshly generated T cells from lymphocyte tissues into the tumour through the CD3 arm. This is suggested from previous data where [89]Zr-labelled ERY974 uptake was observed both in the tumour through the GPC3 arm and in the

**Fig. 7 | ERY974 sensitises MKN45 tumours to capecitabine by increasing conversion of capecitabine to 5′-FU via induction of TP expression. a** GSVA enrichment scores for the GO terms apoptosis, mitotic cell cycle and DNA replication calculated from MKN45 RNAseq data on day 14 ($n = 6$). In the boxplot, centre lines, box limits and whiskers show median values, upper and lower quartiles, minimum and maximum values within quartiles, respectively. Statistical analysis was conducted using two-tailed unpaired *t*-tests (*$P < 0.05$, **$P < 0.01$, ***$P < 0.001$, and n.s. not significant). Exact *p* values of vehicle versus capecitabine, vehicle versus combination, and capecitabine versus combination for apoptosis are $2.56 \times 10^{-1}$, $7.40 \times 10^{-5}$ and $2.45 \times 10^{-3}$, respectively; and for mitotic cell cycle are $2.73 \times 10^{-2}$, $1.45 \times 10^{-5}$ and $1.24 \times 10^{-2}$, respectively; and for DNA replication are $3.02 \times 10^{-2}$, $1.05 \times 10^{-5}$ and $1.00 \times 10^{-1}$, respectively. **b** Western blot analysis for TP expression in MKN45 tumours ($n = 3$). **c** Gene expression of *TYMP* in MKN45 tumours on day 14. Data were presented as the mean ± SD ($n = 6$). Statistical

significance was conducted using the edgeR glmQLSTest (***$P < 0.001$). Exact *p* value is $1.32 \times 10^{-7}$. **d** TDCC of ERY974 in MKN45. Cytotoxicity is presented as the mean ($n = 3$, E:T = 10:1). **e** Western blot analysis for TP expression in MKN45 cells after TDCC (ERY974: 0.001, 0.1, 10 μg/mL) ($n = 1$). **f** IFNγ and TNFα levels in the culture medium of the TDCC assay ($n = 3$). Data were presented as the mean ± SD. **g** Western blot analysis for TP expression in MKN45 cells treated with recombinant IFNγ and TNFα with 0.1, 1, 10, and 100 ng/mL for 24 h ($n = 1$). **h** qRT-PCR for *TYMP* expression in MKN45 cells treated with recombinant IFNγ and TNFα at 100 ng/mL for 24 h ($n = 3$). *TYMP* expression is normalised by *GAPDH* expression. Data were presented as the mean ± SD. **i, j** Cell growth inhibition assay of MKN45 cells treated with 5′-DFUR with or without IFNγ (**i**) and TNFα (**j**) ($n = 6$). Phase-contrast images were captured every 4 h, and cell confluency was automatically calculated. Data were presented as the mean ± SD. Source data are provided as a Source Data file.

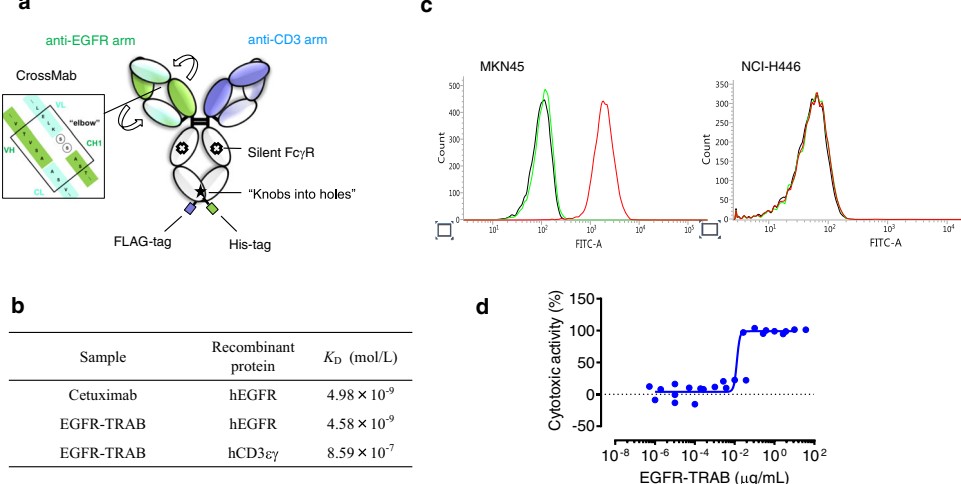

**Fig. 8 | EGFR-TRAB shows TDCC in MKN45. a** Schematic illustration of EGFR-TRAB. Strong and faint green indicates the heavy chain and light chain of the EGFR arm, respectively. Strong and faint purple indicates the heavy and light chain of the CD3 arm, respectively. **b** SPR analysis of EGFR-TRAB. $K_D$ represents the dissociation constant. **c** Cell surface expression of EGFR in MKN45 and NCI-H446 using flow

cytometry with anti-EGFR antibody ($n = 1$). Black line indicates a shift with no antibody; green line indicates a shift with isotype control; red line indicates a shift with an anti-EGFR antibody. **d** TDCC of EGFR-TRAB in MKN45. Cytotoxic activity is presented as the mean ($n = 3$, E:T = 10:1). Source data are provided as a Source Data file.

lymphoid tissues through the CD3 arm in a huNOG model[35]. Exploitation of labelled T cells in T cell-injected models might help to further understand the fate of residual T cells in tumour for future study.

Interestingly, we found that ERY974 promoted the chemotherapy-induced inhibition of cell division. Moreover, we demonstrated that ERY974-induced cytokines, such as IFNγ and TNFα secreted from T cells activated by ERY974 induced TP expression in tumours (Fig. 7c–e). We also showed that induced TP promoted the conversion of capecitabine to its active form at the tumour site, resulting in a synergistic antitumour effect. Collectively, our findings show that chemotherapy promotes ERY974 antitumour activity, and vice versa. Indeed, it has been clinically proven that the combination of pembrolizumab with chemotherapy is more effective than chemotherapy alone in treating non-small cell lung carcinoma[36]. Several clinical trials testing ICIs in combination with chemotherapy to treat various tumours are currently underway; however, detailed mechanisms have not yet been elucidated using clinical samples, and our findings may help explain this combination effect in patients.

In addition to chemotherapy, other types of drugs can be considered for combination therapy with ERY974, including ICIs or anti-angiogenic drugs. As ERY974 administration increases the number of PD-1-positive T cells in tumours (Fig. 3b and Supplementary Fig. 5b), combination therapy with ICIs may be a rational strategy. Indeed, a combination of CEA-TCB with anti-PD-L1 enhanced antitumour activity in a preclinical model[37], and clinical trials for CEA-TCB monotherapy and combination with atezolizumab are currently underway in patients

with metastatic colorectal cancer[12,38]. It has been speculated that an anti-angiogenic drug may make the TME suitable for immunotherapy, such as by polarising tumour-associated macrophages from the immunosuppressive M2 phenotype to the immunostimulatory M1 phenotype, thereby increasing T-cell infiltration into the tumour[39]. Atezolizumab in combination with bevacizumab showed superior clinical activity to atezolizumab or sorafenib monotherapy in patients with hepatocellular carcinoma[40,41] or compared with bevacizumab or sunitinib monotherapy in patients with renal cell cancer[42]. Considering the impact of anti-angiogenic drugs on the TME, combination of anti-angiogenic drugs with ERY974 would be a promising strategy.

In the huNOG model, B and T cells are well-differentiated human cell types, while human myeloid cells are quantitatively and functionally not well-differentiated[22]. Nevertheless, we observed that human macrophages, as well as T cells, highly infiltrated tumours in the ERY974 + paclitaxel group (Fig. 4c). We speculate that ERY974 may induce the differentiation of human myeloid cells through the secretion of various human cytokines required for myeloid cell maturation during the activation of T cells. However, in the huNOG model, human antigen-presenting cells may not contribute to cytotoxicity of cytotoxic T lymphocytes due to a non-matched MHC class I between the tumour and lymphocytes. Further analysis is needed to elucidate the function of various immune cells, such as dendritic cells and macrophages except for T cells in ERY974-induced efficacy, to more precisely understand the mode-of-action of ERY974 in patients.

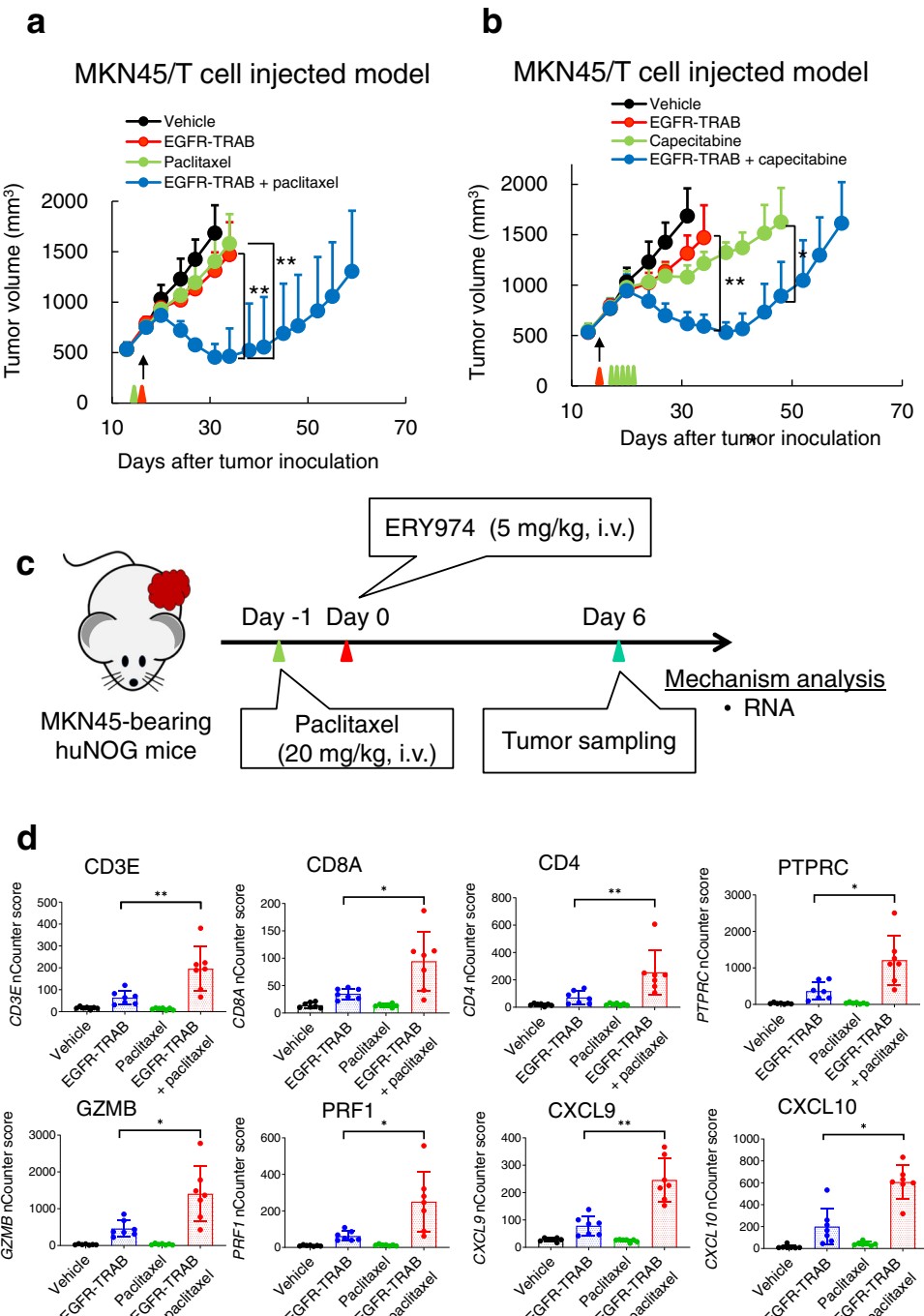

**Fig. 9 | Combination of EGFR-TRAB and chemotherapy shows more efficacy than EGFR-TRAB or chemotherapy alone in non-inflamed MKN45 tumours.**
**a** Antitumour efficacy of EGFR-TRAB combined with paclitaxel in MKN45 tumours in T cell-injected model. EGFR-TRAB (5 mg/kg) and paclitaxel (20 mg/kg) were administered. Red and green arrows indicate the timing of EGFR-TRAB and paclitaxel administration, respectively. Black arrow indicates the timing of T cell administration. Tumour volumes are presented as the mean ± SD (*n* = 5). Statistical significance was evaluated using the Wilcoxon chi-square test (**$P < 0.01$). Exact *p* values of ERY974 versus combination and capecitabine versus combination on day 34 are both 0.0090. **b** Antitumour efficacy of EGFR-TRAB combined with capecitabine in MKN45 tumours in T cell-injected model. EGFR-TRAB (5 mg/kg) and capecitabine (359 mg/kg) were administered. Red and green arrows indicate the timing of EGFR-TRAB and capecitabine administration, respectively. Black arrow indicates the timing of T cell administration. Tumour volumes are presented as the

mean ± SD (*n* = 5). During study, one mouse in the capecitabine group was sacrificed on day 24 due to toxicity. Statistical significance was evaluated using the Wilcoxon chi-square test (**$P < 0.01$, *$P < 0.05$). Exact *p* values of ERY974 versus combination on day 34 is 0.0090, and *p* value of capecitabine versus combination on day 48 is 0.0143. **c** Schematic illustrating the experimental setup for RNA analysis using nCounter (*n* = 7) on day 6. **d** mRNA levels of representative genes for T cell marker, T cell activation marker and chemokine in MKN45 tumours in huNOG model using nCounter data of day 6 samples (*n* = 7). The nCounter scores were compared among each group. Data were presented as the mean ± SD. Statistical significance was conducted using the two-tailed unpaired *t*-test (*$P < 0.05$, **$P < 0.01$, and n.s. no significance). Exact *p* values of ERY974 versus combination for *CD3E, CD8A, CD4, PTPRC, GZMB, PRF1 CXCL9* and *CXCL10* are $8.11 \times 10^{-3}$, $3.39 \times 10^{-2}$, $8.22 \times 10^{-3}$, $1.20 \times 10^{-2}$, $1.00 \times 10^{-2}$, $1.09 \times 10^{-2}$, $1.74 \times 10^{-3}$ and $1.16 \times 10^{-2}$, respectively. Source data are provided as a Source Data file.

Our findings help uncover the mechanisms underlying the synergistic antitumour effects observed between ERY974 and chemotherapy drugs. We propose that combination therapy with ERY974 and chemotherapy may be an effective strategy for patients with non-inflamed tumours and those who are resistant to existing therapies.

## Methods

### Ethics statement
The protocols of all animal studies were approved by the Institutional Animal Care and Use Committee (IACUC) of Chugai Pharmaceutical Co., Ltd. The approved protocol numbers are 15-002, 15-129, 15-213, 15-310, 16-028, 18-505 and 20-289. All animal studies were conducted in the animal facility accredited by the Association for Assessment and Accreditation of Laboratory Animal Care (AAALAC). Housing conditions are 20–26 °C temperature, 30–60% humidity, and 12 h intervals of light and dark. All studies using human samples, e.g. cord blood and PBMCs, were conducted according to the policy of the Chugai Ethical Committee. The approved protocol numbers are 10-427, 10-517 and 10-616.

### Cell lines
The human gastric cancer MKN45 cell lines were purchased from the Japanese Collection of Research Bioresources Cell Bank (JCRB; Osaka, Japan, JCRB0254). The human gastric cancer cell line, MKN74 and human lung cancer cell line, PC10 were purchased from Immuno-Biological Laboratories (IBL; Gunma, Japan, currently discontinued). The lung cancer cell line, NCI-H446 was purchased from the American Type Culture Collection (ATCC; Manassas, VA, USA, HTB-171). The liver cancer cell line, SK-HEP-1was purchased from the Memorial Sloan-Kettering Cancer Center through ATCC (HTB-52). SK-pca13a was engineered to overexpress GPC3 in SK-HEP-1 cells. All cell lines were cultured according to manufacturers' instructions. No further authentication was conducted.

### Reagents
ERY974 was prepared by Chugai Pharmaceutical., Co. Ltd (Tokyo, Japan) as previously described in refs. 15, 43. The transfer of ERY974 to the third parties is restricted as clinical trials are ongoing. Paclitaxel and cisplatin were purchased from FUJIFILM Wako Pure Chemical Corporation (Osaka, Japan) and Bristol-Myers Squibb Co. (New York, NY, USA), respectively. Capecitabine used for the ERY974 combination study was produced by Chugai Pharmaceutical., Co. Ltd, and also purchased from Tokyo Chemical Industry Co., Ltd. (Tokyo, Japan). For EGFR-TRAB combination study. EGFR-TRAB was prepared by Chugai Pharmaceutical., Co. Ltd using anti-CD3ε fragment antigen-binding (Fab) of ERY974 and anti-EGFR Fab of cetuximab. The knobs into holes[44] and CrossMab techniques[30] were applied. For purification, C-terminal FLAG-tag and His-tag were added to the heavy chains of the anti-CD3 arm and anti-EGFR arm, respectively. EGFR-TRAB was expressed in FreeStyle 293-F (Thermo Fisher Scientific, Waltham, MA, USA), and purified by three steps of column chromatography consisting of two affinity purification steps using anti-FLAG M2 affinity agarose gel (Sigma-Aldrich, St. Louis, MO, USA) and Ni Sepharose High Performance (Cytiva, Marlborough, MA, USA), followed by gel permeation chromatography using HiLoad 26/600 Superdex 200 pg (Cytiva).

### In vivo efficacy study using the huNOG model
For establishing the humanised NOG model, female NOD/Shi-scid/IL-2Rγnull (NOG) mice aged 6 weeks were purchased from CLEA Japan, Inc. (Tokyo, Japan). After an acclimatisation of 3 weeks, mice were irradiated (2.5 Gy; MBR-1520R-3; Hitachi Power Solutions Co. Ltd., Ibaraki, Japan) 1 day before transplantation of human CD34+ cells (AllCells, Alameda, CA, USA) via intravenous injection ($1 \times 10^5$ cells/mouse). After approximately three months, $1 \times 10^7$ cells of each cell line were injected subcutaneously into the right flank of huNOG mice with Matrigel (Corning, Corning, NY, USA). When the tumours were established, agents were administered alone or in combination, as described below.

In the combination study with paclitaxel or cisplatin, 20 mg/kg paclitaxel or 7.5 mg/kg cisplatin was intravenously administered once 1 day before the administration of 1 mg/kg ERY974. In the combination study with capecitabine, 359 mg/kg capecitabine was orally administered for 5 consecutive days starting one day before the administration of 5 mg/kg ERY974. Tumour volume (TV) was calculated according to the formula: TV (mm$^3$) = ab$^2$/2, where a is the length of the tumour and b is the width of the tumour. Tumour growth inhibition (TGI) was calculated according to the formula: TGI (%) = [1 − (T − T0)/(C − C0)] × 100, where T and T0 are the tumour volumes of drug-treated groups on a specific day and the initial day, respectively, and C and C0 are the tumour volumes of the vehicle group on a specific day and the initial day, respectively. Measurement of tumour volume and TGI calculations were conducted using the Chugai Antitumour Evaluation System (ANTES), and figures were drawn using Microsoft Excel 2008 (Microsoft, Redmond, WA, USA). According to the protocol approved by the IACUC of Chugai Pharmaceutical Co., Ltd., we completed tumour size measurement in each group on the last day of measurement when at least one mouse fulfilled our criteria for euthanasia, which was when the tumour size exceeded ~2000 mm$^3$. In a few instances this limit has been exceeded and tumour growth followed for few extra days upon assessment of the health status of the animals by the experimenters and approval of the veterinarians.

### In vivo efficacy study using the human T cell-injected model
In the T cell-injected model, female NOD/ShiJic-scidJcl (NOD/SCID) mice aged 5 weeks were purchased from CLEA Japan, Inc (Tokyo, Japan). After an acclimatisation of 1–3 weeks, $1 \times 10^7$ cells of each cell line were subcutaneously inoculated into the right flank of mice. When the tumour was established, anti-Asialo GM1 solution was administered intraperitoneally. Then, 20 mg/kg paclitaxel, or 7.5 mg/kg cisplatin were intravenously administered. On the next day, in vitro expanded human T cells ($3 \times 10^7$ cells/mouse; Cellular Technology Ltd., Shaker Heights, OH, USA) and 1 or 5 mg/kg of ERY974 or EGFR-TRAB were administered intraperitoneally and intravenously, respectively. In the combination of capecitabine with ERY974 or EGFR-TRAB, capecitabine was orally administered at respective 431 or 359 mg/kg for 5 consecutive days starting on the third day after administration of in vitro expanded human T cells and 5 mg/kg ERY974 or EGFR-TRAB. The TGI calculation, and figure drawing were conducted as described above. The permitted maximum tumour size in studies was described above.

### Tumour collection for biomarker and mechanism analysis
For biomarker analysis of baseline and ERY974-treated tumour samples, PC10, NCI-H446, MKN74 or MKN45 cells were inoculated into huNOG mice, as described above. After the tumour was established, vehicle or 1 mg/kg ERY974 was intravenously administered once. After 3 days, the mice were sacrificed (n = 3), and the tumours were collected for RNAseq and histopathological analysis.

To analyze the mechanism of combination therapy utilising the huNOG/NCI-H446 model, huNOG mice inoculated with NCI-H446 tumours were intravenously injected with 20 mg/kg paclitaxel or 7.5 mg/kg cisplatin 1 day before the administration of 5 mg/kg ERY974 in the combination group. In the monotherapy group, each agent was administered as described above. Then, tumours were collected on days 2 and/or 6 after ERY974 administration and used for TIL, RNA and histopathological analyses. For the huNOG/MKN45 model, huNOG mice inoculated with MKN45 tumours were orally administered 359 mg/kg capecitabine for 4 (day 3 samples) or 5 (day 7 and 14 samples) consecutive days, starting 1 day before the administration of 5 mg/kg ERY974. In the monotherapy group, each agent was

administered as described above. Then, the tumours were collected on days 3, 7 and 14 after ERY974 administration and used for RNA and western blot analyses. For the mechanism analysis of the combination of EGFR-TRAB with paclitaxel, huNOG mice inoculated with MKN45 tumours were administered with 20 mg/kg paclitaxel one day before the administration of 5 mg/kg EGFR-TRAB in the combination group. In the monotherapy group, each agent was administered as described above. Then, tumours were collected on day 6 after EGFR-TRAB administration and extracted RNA was used for nCounter analysis.

## Flow cytometry to determine cell surface expression of GPC3 and EGFR

GPC3 or EGFR ABC was determined using QIFIKIT (DAKO, Glostrup, Denmark) according to the manufacturer's instructions. Cultured PC10, NCI-H446, MKN74 or MKN45, cells were treated with accutase (Nacalai Tesque, Kyoto, Japan), after which $5 \times 10^5$ cells were placed in MACS buffer [consisting of autoMACS Rinsing Solution (Miltenyi Biotech, Bergisch Gladbach, Germany) and MACS BSA Stock Solution (Miltenyi Biotech)] and treated with anti-GPC3 mouse monoclonal antibody (in-house preparation)[45], anti-EGFR monoclonal antibody (Biolegend, San Diego, CA, USA), or isotype control at 20 µg/mL. The cells were then incubated for 30 min at 4 °C and washed with MACS buffer, followed by secondary antibody (included in QIFIKIT, 1:50) incubation for a further 30 min at 4 °C. After washing with MACS buffer, cells were analyzed with FACSLyric (BD, Franklin Lakes, NJ, USA). ABC was calculated using the calibration curve obtained using the calibration beads in the kit.

## Tumour lymphocyte analysis

Tumour samples were harvested from recipient mice, minced with scissors, and dissociated using the human tumour dissociation kit (Miltenyi Biotech) and gentle MACS dissociator (Miltenyi Biotech). Single cells were incubated with ACK lysis buffer (Invitrogen, Carlsbad, CA, USA) for 5 min at 4 °C, followed by incubation with cell surface protein-specific antibodies for 30 min at 4 °C. Then, cells were fixed and permeabilized with BD Cytofix/Cytoperm™ Kit (BD Biosciences) and incubated with intracellular protein-specific antibodies for 30 min at 4 °C (Supplementary Table 1). The samples were resuspended in autoMACS Rinsing Solution with MACS buffer, followed by analysis with Fortessa-X20 (BD Biosciences) and data analysis using FlowJo v10.2 (Tree Star Inc., Ashland, OR, USA). Figures were drawn using GraphPad Prism 8.4.3 (GraphPad Software, San Diego, CA, USA).

## Transcriptome analysis

Tumour samples were submerged in RNAlator (Thermo Fisher Scientific) overnight at 4 °C and then stored at −80 °C. Total RNA was extracted from tumours using the miRNeasy Mini Kit (Qiagen, Hilden, Germany). Comprehensive RNA expression analysis was performed with the nCounter PanCancer Immune Profiling Panel (NanoString Technologies, Seattle, WA, USA) using the Prep Station and Digital Analyzer of the nCounter analyzing system (NanoString Technologies) with 100 µg total RNA. RNAseq was conducted at TaKaRa Bio Inc. (Shiga, Japan). The RNAseq reads were mapped to human (GRCh38) and mouse (GRCm38) transcripts simultaneously using RefSeq (Release 62; http://www.ncbi.nlm.nih.gov/refseq/) using bowtie 1.1.2 (maintained by Johns Hopkins University). Reads mapped to the transcripts of both organisms were excluded, followed by the calculation of fragments per kilobase of transcript per million reads mapped (FPKM) using RSEM v1.2.31(The GNU General Public License)[46] parameters as -n 2 -e 99999999 -l 25 -I 1 -S -X 1000 -a -m 200. Figures were drawn using GraphPad Prism 8.4.3.

## Differential expression and gene set analysis

Differential expression analysis was performed using edgeR with the expected counts of each transcript estimated by RSEM. Differentially expressed genes were selected based on two criteria: (i) false discovery rate <0.01 and (ii) absolute fold change >2[47]. Gene set enrichment analysis of differentially expressed genes was performed using Fisher's exact test with the R package clusterProfiler[48].

## Gene signature calculation

Genes belonging to the GO terms apoptosis (GO:0097190), mitotic cell cycle (GO:0000278) and DNA replication (GO:0006260) were selected with the R package AnnotationDBI and their signature scores calculated using GSVA[49]. T cell signature genes used in heatmaps were quoted from a previous study[50].

## Histopathological analysis

For the biomarker analysis of the different tumour models of PC10, NCI-H446, MKN74 and MKN45, the tumours were fixed with 4% PFA (paraformaldehyde) for 24 h and then embedded by the AMeX method. Sliced sections were stained using HE staining and IHC with anti-human CD3 (clone EP449E, Abcam, Cambridge, UK, 1 µg/mL). CD3 IHC was conducted as follows: primary antibodies were applied after antigen retrieval by microwave heating in target retrieval solution, pH 6 (Agilent Technologies, Santa Clara, CA, USA). A labelled polymer reagent (EnVision+ Single Reagents, HRP. Rabbit, K4003, Agilent) was applied as the secondary antibody, and the reaction was visualised using 3, 3′-diaminobenzidine solution (FUJIFILM Wako Pure Chemical Corp). The slides were counterstained with haematoxylin and observed under a light microscope. For quantification of CD3-positive cells, an immune cell count algorithm (Immune cell v1.3, Indica Labs, Albuquerque, NM) in an imaging software (HALO v3.0.311.266, Indica labs, B Albuquerque, NM, USA) was applied. The tumour area was annotated by a pathologist, and cell density (cells per area) was calculated for each tissue. For the mechanism analysis of the ERY974 + paclitaxel combination, isolated tumours were fixed in 10% neutral-buffered formalin, followed by embedding in paraffin. Paraffin-embedded specimens were sectioned and stained with HE. IHC staining for human CD3 and CD68 was also performed; samples were incubated with the primary antibodies mouse anti-human CD3 (clone F7.2.38; DAKO, Glostrup, Denmark, 2.5 µg/ml) and anti-human CD68 (clone PG-M1; DAKO, 1.0 µg/ml) in 0.01 M citrate buffer (pH 6.0) after heating the samples in a microwave. A labelled polymer reagent (EnVision+ Single Reagents, HRP. Rabbit; K4003; Agilent) was applied as the secondary antibody, and the reaction was visualised using 3, 3′-diaminobenzidine solution (FUJIFILM Wako Pure Chemical Corporation). Finally, the slides were counterstained with haematoxylin and imaged under a light microscope.

## TDCC assay

The TDCC assay was performed with the MKN45 or SK-pca13a cell lines as target cells and peripheral blood mononuclear cells (PBMCs) as effector cells at an effector-to-target (E:T) ratio of 10:1. SK-pca13a was engineered to express moderate GPC3 levels in SK-HEP-1, a GPC3-negative cell line. Target cells ($1 \times 10^4$/well) were pre-incubated for 24 h and then various concentrations of ERY974 or EGFR-TRAB and human PBMCs ($1 \times 10^5$ / well) were added. Analysis was then performed using the xCELLigence Real-time Cell Analysis system (ACEA Bioscience, Santa Clara, CA, USA)[15] in a $CO_2$ incubator for 72 h. TDCC (%) was calculated used equation (1).

$$TDCC(\%) = [(A − B)/(A − A0)] \times 100 \qquad (1)$$

A is the mean cell index of wells without antibody, and B is the mean cell index of wells with antibody, and A0 is the mean cell index of wells without antibody just before antibody and PBMCs are added. The data analysis was conducted using Microsoft Excel 2008 and figures were drawn using GraphPad Prism 8.4.3.

## Binding to hCD3 and hGPC3 by enzyme-linked immunosorbent assay

FITC labelling of ERY974 was conducted using the Fluorescein Labelling Kit-NH$_2$ (Dojindo Laboratories, Kumamoto, Japan) according to the manufacturer's instructions. The fluorescence intensity of FITC-labelled ERY974 was measured using a microplate reader (SPECTRAmax M2e; Molecular Devices, San Jose, CA, USA) at excitation (Ex) and emission (Em) wavelengths of 485 and 538 nm, respectively. A total of 100 ng hGPC3 or hCD3e peptides (in-house preparation) were immobilised in 96-well plates. After washing with TBS-T, 100 µL non-labelled or FITC-labelled ERY974 solution was added in duplicate at varying concentrations of 8–512 ng/mL for hCD3 and 0.5–32 ng/mL for hGPC3, followed by incubation for 1 h at room temperature. The samples were then washed and sequentially incubated for 1 h at room temperature with secondary antibody anti-human kappa light chain goat IgG-biotin (Immuno-Biological Laboratories, Gunma, Japan: 0.5 µg/mL), streptavidin-poly HRP (Stereospecific Detection Technologies, Baesweiler, Germany), and TMB super sensitive HRP microwell substrate (SurModics, Eden Prairie, MN, USA). Washing was performed between each incubation step. After final washing, the absorbance at 450 nm was measured with a microplate reader. The data analysis and figure drawing were conducted using Microsoft Excel 2008.

## Distribution study of FITC-labelled ERY974

NCI-H446 tumour-bearing huNOG mice were intravenously administered with vehicle solutions or 20 mg/kg paclitaxel on day 0, and the next day 5 mg/kg FITC-labelled ERY974 was administered in the monotherapy and combination groups. Tumour and blood samples were collected at 24, 72 and 144 h after administration of FITC-labelled ERY974. Tumour tissues excised from each animal were weighed before measuring their fluorescence intensity. Tumour pieces were homogenised, and cell lysates were prepared, after which fluorescence intensity was measured with a microplate reader at Ex 485 nm and Em 538 nm. FITC-labelled ERY974 level in tumour tissues was calculated using the calibration curve, and the resulting value is shown as the concentration of FITC-labelled ERY974 per g tumour (µg eq./g tumour). The data analysis was conducted using Microsoft Excel 2008 and figures were drawn using GraphPad Prism 8.4.3.

## Western blotting of TP in MKN45 tumour samples

Xenograft tumours were cut into small pieces and homogenised using a Multi-Beads Shocker Cell Disruptor (Yasui Kikai, Osaka, Japan). Then, the homogenised tumour cells were lysed with cell lysis buffer [100 mmol/L Tris-HCl (pH 7.5), 150 mmol/L NaCl, 5 mmol/L EDTA, 10% glycerol, 1% Triton X-100, protease inhibitor (Roche Diagnostics, Basel, Switzerland), phosphatase inhibitor (Roche Diagnostics)]. Protein concentrations were determined using a Protein Assay Kit (Bio-Rad Laboratories, Hercules, CA, USA). Then, 50 µg lysate was used for SDS-PAGE. Resolved proteins were transferred onto nitrocellulose membranes using Trans-Blot Turbo (Bio-Rad) and then membranes were probed with anti-thymidine phosphorylase (TP)(1:1000) (Cell Signaling Technology, Danvers, MA, USA) and GAPDH (1:1000) (Cell Signaling Technology). Next, membranes were washed with Tris-buffered saline containing 0.05% Tween-20 (TBS-T) and probed with anti-rabbit IgG-specific Alexa 680-conjugated secondary antibody (Thermo Fisher Scientific, 1:24,000), or anti-mouse IgG -specific Alexa 800-conjugated secondary antibody (Thermo Fisher Scientific, 1:24,000). Finally, membranes were washed with TBS-T and then scanned and analyzed using the Odyssey Infrared Imaging System (LI-COR Bioscience, Lincoln, NE, USA).

## Induced expression of TP in MKN45 cells in vitro

MKN45 cells ($1 \times 10^6$) were cultured in a T75 flask; the next day, $1 \times 10^7$ PBMCs and 0.001, 0.1 or 10 µg/mL ERY974 were added. After 24 h, MKN45 cells were collected by extensive washing. MKN45 cell lysates

were then prepared and western blotting for TP was performed as described above. As for TP induction with recombinant cytokines, MKN45 cells ($3 \times 10^6$) were treated with 0.1, 1, 10 and 100 ng/mL IFNγ (R&D Systems, Minneapolis, MN, USA) or TNFα (R&D Systems) for 24 h. For PC10, NCI-H446 and MKN74, only 100 ng/mL of IFNγ or TNFα were tested. Cell lysate preparation and western blotting were performed as described above using antibodies for GPC3 (prepared in-house, 1 µg/mL) and STAT1 (Cell Signaling Technology, 1:1000) in addition to TP and GAPDH. For qRT-PCR analysis, cells ($5 \times 10^5$) were treated with IFNγ or TNFα for 24 h and total RNA was extracted with the RNeasy Mini Kit (Qiagen), followed by cDNA synthesis using SuperScript III First-Strand Synthesis System (Thermo Fisher Scientific). The qRT-PCR analysis was conducted with SYBR Green Master Mix (Thermo Fisher Scientific) and primers (Sigma-Aldrich)for TP (Forward: 5′-GCTGGAGTCTATTCCTGGATTC-3′, Reverse: 5′-ACTGAGAATGGAGGCTGTGATG-3′) and GAPDH (Forward: 5′-CCCATCACCATCTTCCAGGAGCGA-3′, Reverse: 5′-GCCTTCTCCATGGTGGTGAAGAC-3′) using QuantStudio 12 K Flex (Thermo Fisher Scientific). The data analysis was conducted using Microsoft Excel 2008 and figures were drawn using GraphPad Prism 8.4.3.

## Measurement of cytokine levels

TDCC was conducted as described above. After 24 h, the culture medium was collected. Human cytokine levels were measured using the MAGPIX ®xPONENT®4.2 system (Merck, Kenilworth, NJ, USA) with the Human Cytokine Magnetic 10-Plex Panel (Thermo Fisher Scientific) according to the manufacturer's instructions. The data analysis was conducted using Microsoft Excel 2008 and figures were drawn using GraphPad Prism 8.4.3.

## Cell growth inhibition assay

MKN45 cells ($1 \times 10^4$) in a 96-well plate and 4 µM 5′-DFUR (Sigma-Aldrich), 1 ng/mL IFNγ (R&D systems) or 1 ng/mL TNFα (R&D systems) were added with each agent alone or in combination with chemotherapy. Cell confluency was monitored using the Incucyte Zoom System (Sartorius AG, Göttingen, Germany) according to manufacturer's instructions. Figure drawing were conducted using Microsoft Excel 2008.

## In vitro signature analysis

MKN45 cells ($1 \times 10^4$) were plated in a six-well plate, followed by the addition of 1 µM 5′-DFUR (Sigma-Aldrich) and/or 1 ng/mL IFNγ (R&D systems). After 24 h, RNA was extracted using the RNeasy Mini Kit (Qiagen). For comprehensive RNA expression analysis, 100 ng total RNA was analyzed with nCounter (NanoString) using the PanCancer Pathway Panel (NanoString). Data analysis was performed using nSolver v4.0 (NanoString). Figures were drawn using GraphPad Prism 8.4.3.

## SPR analysis

SPR analysis of EGFR-TRAB to recombinant hEGFR (ACRO Biosystems, Newark, DE, USA) or hCD3εγ (prepared in-house) was measured using a Biacore T200 system (Cytiva). Multicycle kinetics were analyzed at 37 °C. EGFR-TRAB or cetuximab was captured onto a Biacore CM4 sensor surface immobilised with protein A/G (Themo Fisher Scientific). Recombinant biotinylated human EGFR protein, His, Avitag, or recombinant human CD3εγ at varying concentrations (6.25–100 nM for hEGFR or 75–1200 nM for hCD3εγ) were passed over the chip.

Kinetics parameters were determined by fitting sensorgrams with a 1:1 binding model using Biacore T200 Evaluation Software Version 2.0 (Cytiva).

## Statistical analysis and reproducibility

For the antitumour efficacy and FITC-labelled ERY974 distribution studies, statistical analysis was conducted using Wilcoxon rank-sum chi-square tests in the JMP version 11. TIL data analysis was conducted

using JMP version 11 software using two-tailed unpaired *t*-tests. For RNA data analysis, R version 3.6 (R Foundation for Statistical Computing, Vienna, Austria) was used. For RNAseq gene expression, statistical analysis was conducted using edgeR glmQLFTest. For GVSH enrichment score, statistical analysis was conducted using R base functions by a two-tailed unpaired *t*-test. For nCounter gene expression, statistical analysis was conducted using the muticomp Dunnet test and two-tailed unpaired *t*-test. All the data is repeated more than twice, and similar results were obtained.

### Reporting summary
Further information on research design is available in the Nature Research Reporting Summary linked to this article.

## Data availability
The RNAseq data have been deposited to the GEO (Gene Expression Omnibus) with accession number: GSE211373, GSE211374, GSE211512 and GSE211514.

The RNAseq and nCounter data are also available in the Dryad database [https://doi.org/10.5061/dryad.kwh70rz4q].

For RNAseq data analysis, the reference sequence of human(GRCh38) and mouse (GRCm38) transcripts are referred from http://www.ncbi.nlm.nih.gov/refseq/.

Source data are provided with this manuscript. The remaining data are available within the Article, Supplementary Information or Source Data file. Source data are provided with this paper.

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

## Acknowledgements

We thank S. Kishishita, A. Kato, H. Mutoh and M. Hasegawa from Chugai Pharmaceutical Co., Ltd. for their support and advice regarding this study. We also thank M. Noguchi from Chugai Pharmaceutical Co., Ltd for assistance with experiments. This study was funded by Chugai Pharmaceutical Co., Ltd.

## Author contributions

Y. Sano, Y.A., T.T., Y. Kayukawa, J.S., E.F., J.A., Y. Kinoshita, Y. Sakamoto., A.Y., Y.M., Y. Sato and C.T.-S. designed the study. Y. Sano, Y.A., T.T., Y. Kayukawa, J.S., E.F., J.A., Y. Kinoshita, A.Y., Y.M., Y. Sato and C.T.-S. conducted the experiments. Y. Sano, Y.N. and T.M analyzed the data. Y. Sano, J.S., E.F., J.A., Y. Kayukawa, Y.N., Y. Sato, and C.T.-S. wrote the manuscript. T.I., T.T., T.K. and M.E. provided suggestions. All authors reviewed the manuscript.

## Competing interests

All authors are employees of Chugai Pharmaceutical Co., Ltd, which is involved in the research and development of medicines. This study was funded by Chugai Pharmaceutical Co., Ltd. The patent application (WO2017159287: Cell injury inducing therapeutic drug for use in cancer therapy) related to these findings was published by Y.A., Y. Kinoshita., T.T., T.I., M.E. and Y. Sano.
