## [Peer Review File · Nature Communications]

Reviewers' Comments:

Reviewer #1:

Remarks to the Author:

Sano et al. examines the antitumor activity of ERY974, a humanized, bispecific GPC3-CD3 T cell redirecting antibody in subcutaneous xenograft solid tumor models with the objective of understanding the effect of ERY974 administered in combination with chemotherapy. They first show that in non-inflamed tumors (NCI-H446 and MKN45) do not benefit from a single 5mg/kg injection of ERY974 therapy and this finding seems independent of GPC3 expression density on tumor cells. Next, they show in humanized and T cells injected NOG mice bearing NCI-H446 xenografts that combining ERY974 with paclitaxel, cisplatin or capecitabine decreases tumor burden in these mice. They show that T cell infiltration in paclitaxel + ERY974 treated mice is increased on Day 6 and suggest that T cells' gene expression is increased for naïve, inhibitory (activation), cytokines, effector, costim, transcription factors and T reg related markers. They find similar T cell infiltration related signatures in a model of MKN45 xenograft model (no tumor burden or survival curve shown) and show that ERY974 sensitizes MKN45 tumor cells to capecitabine by increasing its conversion to 5FU, the active metabolite.

Comments/questions:

Major:

- 1) The conclusion of the paper is that combination of chemotherapy and ERY974 improves antitumor activity; however, only NCI-H446 cell line-based xenograft size related plots are shown (Fig 2). Broad statements cannot be made based on a single cell line. Furthermore, no survival benefit is shown; therefore, it is unclear if the decrease tumor size is durable. Follow up is variable on the graphs (approximately 32-75 days) and it would have been important not to censor the results for some of the plots (i.e. Fig 2e) at early time points. MKN45 based correlative experiments are described (i.e. T cell infiltration, TP expression, RNAseq), but no data is shown related to survival of mice or tumor burden decrease.
- 2) The authors' statement contradict each other, the rationale for the manuscript is that non-inflamed tumors are resistant to ERY974 alone or checkpoint inhibitors, but they state in the introduction that "Notably, ERY974 shows strong efficacy in LLC1/hGPC3, a non-inflamed tumor where ICIs failed to show efficacy¹²".
- 3) A major concern is the lack of discussion / explanation how T cells presumably activated / engaged by ERY974 which in turn should trigger proliferation, can survive while tumor bearing mice are treated with chemotherapy. In huNOG models, T cells are obviously already present and in the T cell injection models, the effector cells are transferred prior to chemo. Since all three chemotherapeutics evaluated in this manuscript interfere with replication, it is important to understand the fate of effector cells. This could be done by tracking experiments when injected T cells express Ffluc and can be tracked long term.
- 4) Authors present TIL related gene expression profiles in tumor bearing mice treated with paclitaxel and capecitabine (Figs 3d and Fig 6c, respectively). TILs' gene expression profile shows gene upregulated in a very wide range of programs including naïve, inhibitory (activation), cytokines, effector, costim, transcription factors and T reg related markers. This doesn't seem to correspond of current understanding of T cell biology. For example, naïve (less differentiated) cells would not upregulate effector molecules related to cytokines and cytolytic proteins. How can the authors explain such broad upregulation of gene expression programs in theory excluding each other?

Minor:

- 1) Rationale for choosing specific chemotherapeutic agents should be described including comparison to human equivalent dosing. Same should be done for ERY974.
- 2) The authors claim "synergistic and reciprocal" effect for the combination; however, the possible increase of antitumor activity by tumor burden is only shown for NCI-H446 while no antitumor effect is shown for MKN45. Increased effect of conversion of capecitabine to 5FU is only shown for MKN45.
- 3) Authors state incorrectly in the introduction that given the mutated IgG4 backbone, antigen-dependent CRS is unlikely; however, it is clearly observed in cell therapies that T cell activation itself can induce CRS.
- 4) IHCs in Fig 4c and Suppl Fig 1 a-c are hard to interpret at resolution provided. The reviewer cannot confirm the findings described.

- 5) Section heading of the results section should state the findings not the description of analysis/assays.
- 6) Fig 2. timing of T cell injection for the corresponding groups should be indicated.
- 7) Suppl Fig 3a: Typo should be corrected: cisplaton->cisplatin.
- 8) Define hPTPRC in the text before using abbreviation.
- 9) Genes should be listed for the rows of the heatmap shown in Fig 6c.
- 10) In methods, SK-HEP-1 is listed as one of the engineered cell lines used, but it is not clear where this cell line was used in the manuscript (not shown in any figs and not found anywhere else in the text).

Reviewer #2:

Remarks to the Author:

Some immunotherapy treatments are known to be very effective against cancer. However, adaptive immune resistance mechanisms can compromise the success of these therapies. Here, the authors focus on strategies to enhance the response of non-inflamed tumors to immunotherapy. They propose to use chemotherapy to restore the response of some non-inflamed tumors to ERY974, a humanized bispecific antibody that has potent anticancer activity against solid tumors (published in Science Translational Medicine in 2017). The authors found that the combined treatment with ERY974 and chemotherapy was highly effective in preventing tumor growth.

General comments:

The work presented is overall interesting and the strength of the study relies on the models used (humanized mouse models). However, it should be noted that:

- 1) the authors have already reported the anticancer effects of their antibody in the STM paper.
- 2) a substantial body of literature has already documented the synergistic effects of chemotherapy and immunomodulation (please refer for instance to PMID: 26872698). The relevance of chemo-immunomodulation has also been discussed in humans, notably in lung cancer. In the present case, the observations are surely interesting, but need to be presented in their proper perspective because of the absence of a major conceptual advance.
- 3) the proposed mechanistic investigations remain descriptive. For instance, it has been reported in mice and humans that some chemotherapies may harbor immunogenic features and others not depending on the cancer model. This has unfortunately not been explored here. It might have been helpful to explore in detail the mechanisms of action of a given combination rather than providing limited information on three different chemotherapies.

In summary, this is an earnest study that has merit. Because of the lack of strong mechanistic insight and the numerous previous studies published in the field (including the one of the authors), my enthusiasm for the present work is however not strong.

Reviewer #3:

Remarks to the Author:

The manuscript by Sano et al describes synergisms between chemotherapy and ERY974 (CD3xGPC3 bispecific antibody) in mouse tumor models demonstrating rational options for combination treatments of non-inflamed tumors, in particular. Mechanisms of action are studied.

The manuscript is highly interesting, well written and provides advances in the thinking about immunotherapy with CD3-based T cell redirection. The anti-tumor activity in the combination studies observed in Fig. 2 (as compared to Fig. 1) I find most impressive.

Comments:

Introduction:

-I encourage the authors to include investigational treatments outside of the Roche family of compounds (such as the CEA-TCB). Examples of CD3-based bispecific that are being tested in non-inflamed tumors, e.g. include AMG160 and TNB 858 in prostate cancer.

-ERY974 is mentioned to be evaluated in a clinical trial. In clinicaltrials.gov, ERY974 however there is only a single ERY974 study shown, with a completion date of August 2019. The authors should update this information and disclose where the antibody is currently being clinically investigated (or not).

-line 70 indicates weekly injection is sufficient to maintain the antibody at sufficient levels in the blood. The manuscript referred to however only shows a mouse study. Where is the conclusion based on? Is the ref 12 not more relevant in this context. What is known/disclosed about PK in humans from the completed clinical study?

Results:

-Figure 1 b. It is not clear to me if the black dots show the data of the tumor in the presence of T cells? Confirm that T cells were injected in both, and the differences observed are only due to the presence or absence of the bispecific T cell engager.

-Did the authors study tumor growth in the absence of T cells as well? Alloreactivity is often challenge in these types of studies. How was this controlled for? Were donors preselected to avoid alloreactivity against the cell lines shown?

-In light of the above, were the in vivo studies shown performed with T cells derived from multiple donors with similar results? In other words, can the authors exclude that differences in the 'inflammatory' or 'non-inflammatory' phenotype observed is a result of alloreactivity of the T cells rather than a characteristic of these xenografts? This information should be added as supplementary data.

-NCI-H446 tumors were treated with a combi of ERY974 and paclitaxel and with cisplatin. MKN45 tumors were treated with a combi of ERY974 and capecitabine. It is not clear to me how these tumor and treatments were matched. Were NCI-H446 not treated with capecitabine and visa versa MKN45 not with paclitaxel or cisplatin / ERY974 combi? Or did these treatments not give the desired result? Please elaborate on the rationale of treatment choice and discuss results from combi treatments with other tumors. It would be critical to demonstrate that the combis work more generally and not just in one tumor model.

Discussion:

-line 268 says that IFN and TNF convert capecitabine. However the authors, I believe, mean to say that this effect occurs indirectly through the induction of thymidine phosphorylase expression by the tumor. Please correct. It would be interesting to discuss how general a phenomenon this might be, or could this be restricted to the MKN45 model? Also see my last comment to the results section above.

Reviewer #4:

Remarks to the Author:

In the study "Combination of the T cell-redirecting bispecific antibody ERY974 and chemotherapy reciprocally enhances antitumor activity against non-inflamed tumors" Sano et al. describe their investigation of combination effects of the T cell-redirecting bispecific antibody ERY974 and chemotherapy in non-inflamed tumors using a combination of techniques in cell lines and mouse xenograft models.

They first demonstrate the evidence that high expression of GPC3 is not sufficient to induce tumor regression, as evident by limited effect in NCI-H446 xenografts. Surprisingly, they also demonstrate that high expression of GPC3 is also not necessary for an ERY974 anti-tumor effect, as evident by the tumor growth inhibition in MKN74 xenograft, where MKN74 has low expression of GPC3. They show evidence that immune cell infiltration is predictive for the efficacy of ERY974 monotherapy.

Question: Authors speculate that "that residential myeloid cells may help to increase number of T cells after ERY974 treatment by secreting cytokines, or chemokines". As according to methods,

RNA material was collected from MKN74 xenograft treated with ERY974, I wonder whether it would be possible to check if the author's cytokine mediated T-cell recruitment speculation can be supported by data, by performing RNA-seq of those samples.

The authors further demonstrate an increased anti-tumor effects of combination treatment of ERY974 with chemotherapy in tumors with low infiltration of immune cells (NCI-H446 and MKN45). They further investigate the mechanism of such increased anti-tumor effect. They demonstrate that in NCI-H446 model in a combination treatment group there is an increased evidence of several biomarkers of T-cells cells presence and activity. Through histopathological analysis the authors demonstrated increased presence of immune cells within the tumor in the combination treatment group, suggesting that chemotherapy enhances ERY974 effect by disturbing the tumor structure and allowing more immune cells and more of ERY974 to infiltrate the tumor.

The authors confirmed their finding using RNA from MKN45 xenograft.

The authors also show that not only chemotherapy enhances the effect of ERY974 by allowing more infiltration, but also that ERY974 treatment increases the efficacy of chemotherapy. They show this by RNA analysis of MKN45 xenograft.

Question:

If the authors have the data for both MKN45 and NCI-H446 xenografts, why do they only show data for one of the models in the rest of the manuscript?

To confirm their hypothesis about enhanced chemotherapy effect in combination treatment, the authors examine TP expression in all groups using western blotting, and find that TP expression is increased in combination treatment.

Comment: If the data is available, this finding should be confirmed from RNA data.

Methods:

Comment: Cell line SK-pca13a is not mentioned anywhere in the main text (except figure legend).

Comment: though the text the authors identify the time points where RNA was collected, but they do not specify whether this RNA was subjected to sequencing of Nanostring quantification method. It therefore came as a surprise to me to see that both Nanostring and RNA-seq was used. Stating more clearly in the text where quantification is done by RNA-seq and where is it done by Nanostring would address this comment.

Comment: In the methods section for transcriptome analysis it is not indicated which software was use to align reads to mouse and human genome. Ideally, the authors should provide the name and version of used software, as well as the list of parameters that were used for the analysis (if different from default).

Comment: The sentence describing normalization procedure for RNA-seq reads is confusing. RSEM provides TPM (transcripts per million), which is a current accepted normalization method; FPKM (fragments per kilobase per million) is calculated using a different procedure and tool, and is not advised to be used. From the sentence it is not clear what was done with the data. Please, state clearly whether TPM or FPKM was used. From this sentence it follows that FPKM normalization was applied to TPM values, which should not be done.

Conclusion: I found the study to be very interesting, and overall of good quality (given the comments above are addressed). The evidence of increased anti-tumor effects of combination treatment in mouse models is convincing, though the major drawback in my opinion is use of a single cell line as evidence of the mode of action (the data seems to be available though, but it is not evident from the manuscript, as single cell line is used for each point made by the authors). I am still intrigued by the effect of ERY974 monotherapy MKN74 xenograft, but this seems to be out of scope of this study, as the main focus is the combination therapy and it's mode of action.

My background is in Cancer genomics and bioinformatics analysis, with extensive experience in transcriptomics. I have worked with RNA-seq data and Nanostring data, from human cell lines, mouse models and cancer patients. Therefore my questions might not require manuscript modification and may be addressed by the authors' explanation, but my comments should be addressed in the manuscript.

29th October 2021

Dear reviewers,

Thank you for allowing us to improve our manuscript entitled “Combination of T cell-redirecting bispecific antibody ERY974 and chemotherapy reciprocally enhances efficacy against non-inflamed tumors” with your valuable comments and queries. It is our belief that the manuscript is substantially improved after making the suggested edits. We have worked hard to incorporate your feedback into the revised manuscript and hope that these revisions persuade you to accept our submission.

Following this letter is a point-by-point response to the reviewers’ questions and comments. This revision was made in consultation with all coauthors, and each author has approved the final manuscript.

Sincerely,

Yuji Sano, Ph.D.

Discovery Pharmacology Department

200 Kajiwara, Kamakura, Kanagawa 247-8530, Japan

Tel: +81-467-47-6241

Fax: +81-467-47-2234

Email: sanoyuj@chugai-pharm.co.jp

Comments from Reviewer #1 (Remarks to the Author): with expertise in glypican-3 targeting

Sano et al. examines the antitumor activity of ERY974, a humanized, bispecific GPC3-CD3 T cell redirecting antibody in subcutaneous xenograft solid tumor models with the objective of understanding the effect of ERY974 administered in combination with chemotherapy. They first show that in non-inflamed tumors (NCI-H446 and MKN45) do not benefit from a single 5mg/kg injection of ERY974 therapy and this finding seems independent of GPC3 expression density on tumor cells. Next, they show in humanized and T cells injected NOG mice bearing NCI-H446 xenografts that combining ERY974 with paclitaxel, cisplatin or capecitabine decreases tumor burden in these mice. They show that T cell infiltration in paclitaxel + ERY974 treated mice is increased on Day 6 and suggest that T cells' gene expression is increased for naïve, inhibitory (activation), cytokines, effector, costim, transcription factors and T reg related markers. They find similar T cell infiltration related signatures in a model of MKN45 xenograft model (no tumor burden or survival curve shown) and show that ERY974 sensitizes MKN45 tumor cells to capecitabine by increasing its conversion to 5FU, the active metabolite.

Comments/questions:

Major 1-1 “The conclusion of this study is that the combination of chemotherapy and ERY974 improves antitumor activity; however, only NCI-H446 cell line-based xenograft size related plots are shown (Fig 2). Broad statements cannot be made based on a single cell line. Furthermore, no survival benefit is shown; therefore, it is unclear if the decrease in tumor size is durable. Follow up is variable on the graphs (approximately 32-75 days) and it would have been important not to censor the results for some of the plots (i.e., Fig. 2e) at early time points.”

Thank you for your comment. So far, we have identified NCI-H446(lung) and MKN45 (gastric) as non-inflamed tumor models. We consider hepatocellular carcinoma as the target cancer type of ERY974, followed by lung and gastric cancers, as GPC3 is highly expressed in these types of cancers¹. In Fig. 2, we showed only the representative data of matched combinations of chemotherapy drugs and tumor types in which these chemotherapy drugs are clinically used. For example, we combined ERY974 with paclitaxel or cisplatin in the NCI-H446 (lung) tumor model because cisplatin and/or paclitaxel are frequently used as standard therapies in lung cancer², particularly in

patients with gene alterations negative for *EGFR*, *ALK*, *ROS1*, *BRAF*, *MET*, or *RET*, etc, and we combined ERY974 with capecitabine in the MKN45 (gastric) tumor model because fluorouracil-related drugs including capecitabine is a standard therapy used in gastric cancer³. However, based on your suggestion, we added the data for the combination of ERY974 with paclitaxel or cisplatin in MKN45 and the combination of ERY974 with capecitabine in NCI-H446 (Supplementary Fig.3). It should be noted that cisplatin is sometimes used in combination with fluorouracil-related drugs in gastric cancer treatment as first-line treatment and paclitaxel is also used in gastric cancer as second-line treatment. In most cases of our efficacy studies, the combination of ERY974 with chemotherapy drugs tended to increase efficacy.

Recently, it has become more difficult to conduct efficacy experiments whose endpoint is death in murine preclinical studies, as we follow the International Animal Care and Use Committee (IACUC) policy of Chugai Pharmaceutical Co., Ltd. when conducting animal studies. As per the IACUC policy, mice must be euthanized when tumor weight reaches 10% of body weight to avoid tumor burden, corresponding to approximately 2000 mm³ tumor volume. In the IACUC policy, studies utilizing actual death as an endpoint are strictly regulated and allowed only when there is no alternative method. In our xenograft study, we did not meet this criterion. Therefore, we could not conduct an efficacy study in which death was an endpoint.

As for the timing of the censor, we completed the efficacy study on day 32, as described in Figure 2e (combination of ERY974 with capecitabine in the MKN45 model). Therefore, we cannot conduct statistical analysis at a later point. Instead, in the newly conducted EGFR-TRAB efficacy study, we observed tumor size at longer time points in mice administered with EGFR-TRAB and capecitabine in MKN45 (Fig. 8f), in which tumors eventually showed regrowth around day 40, possibly because capecitabine was insufficient or the MKN45 tumor had difficulty showing a complete response. However, on day 40, the efficacy of the combination therapy was still statistically significant.

Major 1-2 MKN45 based correlative experiments are described (i.e. T cell infiltration, TP expression, RNAseq), but no data is shown related to survival of mice or tumor burden decrease.

The *in vivo* efficacy data of ERY974 combined with capecitabine in MKN45 cells are shown in Fig. 2e-f. Additionally, the *in vivo* efficacy data of EGFR-TRAB combined with capecitabine in MKN45 has been added, as shown in Fig. 8f.

Major 2 The authors' statement contradict each other, the rationale for the manuscript is that non-inflamed tumors are resistant to ERY974 alone or checkpoint inhibitors, but they state in the introduction that "Notably, ERY974 shows strong efficacy in LLC1/hGPC3, a non-inflamed tumor where ICIs failed to show efficacy¹².

Thank you for your comment. Yes, our description in the introduction section may be excessive. Therefore, we revised "strong efficacy" to "significant efficacy". We believe that TRAB monotherapy could show efficacy to some extent even in non-inflamed tumors; however, combination with chemotherapy drugs would increase the efficacy.

Major 3 A major concern is the lack of discussion and explanation of how T cells presumably activated/engaged by ERY974, which in turn should trigger proliferation, can survive when tumor-bearing mice are treated with chemotherapy. In huNOG models, T cells are already present, and in the T cell injection models, the effector cells are transferred prior to chemo. Since all three chemotherapeutics evaluated in this manuscript interfere with replication, it is important to understand the fate of effector cells. This could be done by tracking experiments when injected T cells express Ffluc and can be tracked long term.

This is an important query. As stated by you, chemotherapy drugs can inhibit the proliferation of T cells as well as tumor cells. In contrast, ERY974 induces T cell proliferation as described in our previous publication¹. You have raised an important suggestion; however, due to technical issues, we cannot conduct T cell imaging experiments. We speculate that chemotherapy drugs may initially inhibit T cell proliferation. However, ERY974 can strongly induce the proliferation of residual T cells in tumors, or recruit fresh T cells from lymphoid tissues through its CD3 arm. In our manuscript, *CD3* expression in MKN45 tumor in early timepoint seems lower in ERY974+capecitabine group than in ERY974 group, however in later timepoint of 14 days, sustained *CD3* expression was observed only in combination group (Fig.6b). These data suggest that initial damage of T cells by capecitabine was overcome by ERY974-induced proliferation of T cells by day 14. Furthermore, in our previous study in collaboration with Dr. Elisabeth GE de Vries, a Professor of Medical Oncology at the University Medical Centre Groningen, we found that the uptake of ⁸⁹Zr-labeled ERY974 was observed in both the tumor via its GPC3 arm and the spleen and lymph nodes via its CD3 arm in the huNOG model⁴, suggesting that ERY974 may be recruiting fresh T cells

from lymphoid tissues into the tumor site, thereby counteracting the negative impact of chemotherapy on T cells. Moreover, we administered all chemotherapies (except capecitabine) only once, which may have limited inhibition of T cell proliferation caused by chemotherapies in our experiments. Clinically, the combination of atezolizumab or pembrolizumab with chemotherapy is currently approved for lung cancer treatment^{5,6}, and pembrolizumab plus chemotherapy is also approved for triple-negative breast cancer based on the KEYNOTE-355 results⁷. The detailed mechanism of this combination effect has not been fully elucidated, but as immune checkpoint inhibitors can also induce T cell proliferation, this clinical evidence may partially support our hypothesis. We have added this explanation in the Discussion section.

Major 4 authors present TIL-related gene expression profiles in tumor-bearing mice treated with paclitaxel and capecitabine (Figs 3d and 6c, respectively). TIL gene expression profiles show genes upregulated in a wide range of programs, including naïve, inhibitory (activation), cytokines, effector, costim, transcription factors, and T reg related markers. This does not seem to correspond to the current understanding of T cell biology. For example, naïve (less differentiated) cells do not upregulate effector molecules related to cytokines and cytolytic proteins. How can the authors explain such a broad upregulation of gene expression programs in theory, excluding each other ?

You have raised an important question. We speculate that when ERY974 binds to T cells, it cannot discern the differentiation status of T cells, as ERY974 recognizes T cells through CD3, which is expressed in most T cell subsets. Therefore, ERY974 may induce proliferation of various populations of T cells (e.g., naïve T cells, cytotoxic T cells, memory T cells, and CD4 T cells) irrespective of their differentiation status. As for CD4+ T cells, we previously demonstrated that ERY974 can activate CD4+ T cells to exert cytotoxic activity¹, and our unpublished data shows that CD4+ T cells secrete higher levels of cytokines than CD8+ T cells, which may cause upregulation of cytokine genes upon administration of ERY974 with and without chemotherapy. Furthermore, it has been reported that tebentafusp, an atypical type of TRAB, can utilize T cells at various differentiation statuses, including CD45RA effector memory, central memory, effector memory, and naïve T cells, as effector cells, and cytokines can be secreted from any T cell subtype upon tebentafusp stimulation, although the secretion level of cytokines differs depending on the subtype⁸. The other possibility is that the broad upregulation of gene expression may be caused by indirect or secondary effects. ERY974 first activates CD8+

cytotoxic T cells. Activated T cells may express various cytokines and/or chemokines, which can activate and recruit immune cells other than cytotoxic CD8+ T cells, such as naïve T cells, memory T cells, CD4+ T cells, Tregs, myeloid cells, and B cells into tumor. Consistent with this, flow cytometric analysis of PBMCs from patients administered with tebentafusp showed decreased levels of CXCR3 in all the CD4/8+ subtypes, including effector memory, central memory, stem cell memory, and naïve memory T cells in peripheral blood, suggesting that their infiltration into the tumor is mediated by tebentafusp-induced CXCL10, a CXCR3 ligand⁹, in tumor. We may have observed not only direct ERY974 events but also indirect ERY974 events in our gene expression analysis.

Minor 1 Rationale for choosing specific chemotherapeutic agents should be described including comparison to human equivalent dosing. Same should be done for ERY974.

The rationale for choosing chemotherapeutic drugs is described in the major (1) answer, and we added description in the main text. The clinical regimen of chemotherapy drugs is complicated. All the chemotherapy drugs used in our study have various differing doses and regimens in clinical use and so we cannot use precisely the same doses and regimens as those used clinically. Therefore, in our study, we aimed to demonstrate the mode of action of combination therapy. In clinical scenarios, chemotherapy drugs are normally used at the maximum tolerated dose (MTD). Therefore, we used doses close to the MTD in mice. In fact, we first conducted a dose-escalation study to determine the MTD for some of the chemotherapy drugs before undertaking the combination study. Regarding the ERY974 dose, based on our preclinical studies, we concluded that it is not suitable to consider the clinical dose in a murine efficacy study, as both huNOG and T cell-injected models are immunodeficient and these models do not fully reconstitute the human immune system, creating a gap in the efficacy dose between our murine models and actual clinical use. The efficacy dose calculated from the huNOG model was several hundredfold higher than that calculated from the *in vitro* assay. Based on these data, the first-in-human dose of ERY974 was decided not from a mouse xenograft study, but from the minimum anticipated biological effect level (MABEL) *in vitro* (our internal data) and no observed adverse effect level (NOAEL) in cynomolgus monkey. In fact, in our Phase I trial (NCT02748837), one patient showed partial response when administered with 0.58 µg/kg ERY974, which is much lower than the preclinically determined efficacy dose¹⁰. Similar data were observed for tarlatamab, a half-life extended (HLE) bispecific T-cell engager (BiTE) targeting DLL3. The preclinical model determined that 3 mg/kg

tarlatamab is required to show maximum efficacy *in vivo* against patient-derived xenograft (PDX) tumors in a T cell injected model using NOG mice¹¹. However, in the Phase I dose-escalation study, tarlatamab showed partial response (PR) at 0.3 mg/patient¹², which is around 4.3 µg/kg if patient weight is assumed to be 70 kg, which is several hundredfolds lower than that obtained for the preclinical model. The gap in efficacy dose between mice and patients was also observed in AMG160, an HLE PSMA-targeted BiTE. In the preclinical model, 0.2 mg/kg was required to achieve PR against 22Rv-1 tumor¹³. On the other hand, in patients, AMG160 showed a greater than 50% reduction in the PSA level in blood at 0.03 mg, which corresponds to 0.4 µg/kg, assuming patient weight as 70 kg^{14, 15}. In our previous efficacy studies, we used 1 or 5 mg/kg ERY974, at which the maximum efficacy of ERY974 was expected in our models.

Minor 2 The authors claim “synergistic and reciprocal” effect for the combination; however, the possible increase of antitumor activity by tumor burden is only shown for NCI-H446 while no antitumor effect is shown for MKN45. The increased effect of conversion of capecitabine to 5FU is only shown for MKN45.

We showed antitumor data of MKN45 in Fig. 2e-f (e: combination of ERY974 with capecitabine in T cell injected model, f: combination of ERY974 with capecitabine in huNOG model), and in the newly added Supplementary Fig. S1a-b (a: combination of ERY974 with paclitaxel in T cell-injected model, b: combination of ERY974 with cisplatin in T cell-injected model), and in Fig. 8e-f (e: combination of EGFR-TRAB with paclitaxel in T cell injected model, f: combination of EGFR-TRAB with capecitabine in T cell injected model). As for the effect of capecitabine conversion, we added data showing that *TYMP* gene expression and TP protein synthesis is induced by IFN γ and TNF α in NCI-H446, PC10, and MKN74 (Fig. S9c-d). From these results, we believe that capecitabine conversion to active form by ERY974-induced IFN γ , and TNF α is not limited to MKN45.

Minor 3 authors state incorrectly in the introduction that given the mutated IgG4 backbone, antigen-dependent CRS is unlikely; however, it is clearly observed in cell therapies that T cell activation itself can induce CRS.

Thank you for your comment. We intended to write “antigen-independent CRS”. As indicated by you, it is clear that CRS is one of the major issues regarding CAR-T and TRAB use, especially in catumaxomab, a TRAB targeting EpCAM, which was approved

by the European Medicines Agency (EMA) in 2009 for EpCAM-positive malignant ascites but is currently withdrawn from the market, possibly due to severe toxicity. Catumaxomab is a trifunctional monoclonal antibody consisting of the mouse IgG2a chain recognizing EpCAM, rat IgG2a chain recognizing CD3, and hybrid Fc (mouse and rat) recognizing FcγR-type I, IIa, and III expressed in macrophages and NK. Functional Fc could induce ADCC, which is believed to increase antitumor efficacy¹⁶. However, severe CRS and liver toxicity were observed in patients, which may have been caused by active Fc¹⁷. By writing that sentence in the introduction, we intended to inform the reader that we used mutant Fc to which FcγR cannot bind, thereby potentially mitigating Fc-dependent (or GPC3-independent) CRS, as we recognize that GPC3-dependent toxicity is observed in patients. To mitigate GPC3-dependent CRS, we are now premedicating subjects with tocilizumab (anti-IL6R antibody) in the ERY974 phase I study (JapicCTI-1948).

Minor 4 IHCs in Fig 4c and Suppl Fig 1 a-c are difficult to interpret at the resolution provided. The reviewer cannot confirm the findings described.

We have provided a higher resolution image.

Minor 5 Section heading of the results section should state the findings not the description of analysis/assays.

We have corrected this, as advised.

Minor 6 Fig 2. timing of T cell injection for the corresponding groups should be indicated.

Thank you for your suggestion. We have added the arrow as the timing of T cell injection in all the data using T cell injected model (Fig. 2 a,c,e, Fig.8e,f, Fig. S1c, and Fig. S3a-c), as advised.

Minor 7 Suppl Fig 3a: Typo should be corrected: cisplaton->cisplatin.

Thank you for pointing out. We changed to the right spelling.

Minor 8 Define hTPRC in the text before using abbreviation.

We explained hPPTPRC as human protein tyrosine phosphatase receptor type C (PTPRC) when we first used this term in the manuscript.

Minor 9 Genes should be listed for the rows of the heatmap shown in Fig 6c.

Thank you for pointing out. We added gene names in the heatmap in Fig.6c.

Minor 10 In methods, SK-HEP-1 is listed as one of the engineered cell lines used, but it is not clear where this cell line was used in the manuscript (not shown in any figs and not found anywhere else in the text).

Thank you for pointing out. We used SK-pca13a cell line, which is an engineered cell to express GPC3 in SK-HEP-1 that lacks GPC3 expression, to compare TDCC between original ERY974 and FITC-labelled ERY974 used for distribution study (Fig. 5a). Per your comment, we added the explanation of SK-pca13a in the relevant part.

Comments from Reviewer #2 (Remarks to the Author): with expertise in chemotherapy and immunomodulation

Some immunotherapy treatments are known to be very effective against cancer. However, adaptive immune resistance mechanisms can compromise the success of these therapies. Here, the authors focus on strategies to enhance the response of non-inflamed tumors to immunotherapy. They propose to use chemotherapy to restore the response of some non-inflamed tumors to ERY974, a humanized bispecific antibody that has potent anticancer activity against solid tumors (published in Science Translational Medicine in 2017). The authors found that the combined treatment with ERY974 and chemotherapy was highly effective in preventing tumor growth.

General comments:

The work presented is overall interesting and the strength of the study relies on the models used (humanized mouse models). However, it should be noted that:

- 1. The authors have already reported the anticancer effects of their antibody in the STM paper.***

Thank you for your comment. As you suggested, we have already shown that ERY974 has antitumor activity in different tumor models. However, the previous paper focused only on those models in which ERY974 showed strong efficacy. There were two other models, MKN45 and NCI-H446, in which ERY974 showed moderate efficacy, but these were not covered. Since then, we have been studying the reason why ERY974 displays only moderate efficacy in these tumors and how to increase that efficacy. In the present study, we demonstrated that a non-inflamed tumor microenvironment (TME) prevents TRABs from exhibiting strong efficacy and that this limitation can be overcome by combining TRABs with chemotherapy drugs. We assume that in clinical scenarios, some patients may not benefit from ERY974 due to the non-inflamed TME, despite sufficient GPC3 expression levels. Therefore, elucidation of the reason why ERY974 is unable to show strong efficacy in these specific tumor models and identification of strategies to overcome this issue should be useful in the clinical development of ERY974.

- 2. A substantial body of literature has already documented the synergistic effects of chemotherapy and immunomodulation (please refer for instance to PMID:***

26872698). The relevance of chemo-immunomodulation has also been discussed in humans, notably in lung cancer. In the present case, the observations are certainly interesting, but need to be presented in their proper perspective because of the absence of a major conceptual advance.

Thank you for providing these insights. We believe that the novel discovery in our study is that even if chemotherapy does not induce immunogenic cell death (ICD), the synergistic antitumor efficacy of a combination of TRAB and chemotherapy in non-inflamed tumors is observed and this strategy may be generalizable to wide variety of tumor antigens of TRAB aside from GPC3. Regarding your comment, we understand that depending on the tumor model, specific chemotherapy drugs may cause ICD, which activates the immune system. Unfortunately, we did not observe significant immune activation by chemotherapy drugs alone in our murine models (Fig. 3,4, 6, and Fig. S5). We speculate that this is because huNOG and T cell-injected models are constitutively immunodeficient and lacking functional human DCs, NK cells, and B cells (the latter only in a T cell-injected model). According to the literature, DCs are necessary to induce immune activation through ICD, as DCs are activated by damage-associated molecular patterns (DAMPs) secreted from dying tumor cells, which then trigger the immune system¹⁸. Our xenograft model was not suitable for examining the ICD effect of chemotherapy drugs. Nevertheless, we observed that chemotherapy drugs show synergistic effects when combined with ERY974 in terms of antitumor efficacy and infiltration of T cells, even if ICD does not occur. The mechanism underlying this event is different from the chemo-immunomodulation effect reported in previous studies. We believe that disruption of the physical barrier of tumor architecture by chemotherapy drugs to allow T cells to infiltrate into the center of the tumor may be a key mechanism underlying this combination effect. Furthermore, we showed, for the first time, that chemotherapy drugs enhance the distribution of labeled ERY974 into tumors.

3. The proposed mechanistic investigations were descriptive. For instance, it has been reported in mice and humans that some chemotherapies may harbor immunogenic features, while others do not depend on the cancer model. However, this has not been explored here. It might have been helpful to explore in detail the mechanisms of action of a given combination rather than providing limited information on three different chemotherapies.

Regretfully, immunocompetent humanized mice are not currently available for the

evaluation of TRABs. We expect that we could observe a more drastic effect of combination therapy in the future using immunocompetent humanized mice, whereby the ICD effect could be added to the efficacy findings. We have added a description of this in the Discussion section. The other novel discovery described in our study is that ERY974 and chemotherapy drugs are reciprocally beneficial. Based on our RNA analysis findings (Fig. 7a and Fig. S8), along with the chemotherapy drug's role in enhancing the distribution of labeled ERY974 and T cell infiltration into the tumor, ERY974 also appears to enhance the chemotherapy drug's function of inhibiting cell cycle progression and inducing apoptosis. We have not yet elucidated the underlying mechanisms of every combination. However, for capecitabine, we identified, for the first time, that ERY974-induced TP induction efficiently converts capecitabine to its active form. We believe that this finding is important and may provide a compelling rationale for the clinical use of this combination.

In summary, this is an earnest study that has merit. Because of the lack of strong mechanistic insight and the numerous previous studies published in the field (including the one of the authors), my enthusiasm for the present work is however not strong.

We appreciate your assessment of our work. We think that in the near future, patients resistant to TRABs including ERY974 will begin to clinically emerge, and treatment strategies for such patients will inevitably be needed. Our proposal to combine TRAB with chemotherapy for such patients could be one of the options to overcome resistance. We also clarified not all, but some of the mechanisms of synergy, which might be a rationale for this combination in the future. This strategy could be applicable to the clinical development of most TRABs, not just ERY974. Therefore, we believe that the findings in this paper will be of great importance to investigators as well as to readers of Nature Communications.

Comments from Reviewer #3 (Remarks to the Author): with expertise in bispecific antibodies

The manuscript by Sano et al describes synergisms between chemotherapy and ERY974 (CD3xGPC3 bispecific antibody) in mouse tumor models demonstrating rational options for combination treatments of non-inflamed tumors, in particular. Mechanisms of action are studied.

The manuscript is highly interesting, well written and provides advances in the thinking about immunotherapy with CD3-based T cell redirection. The anti-tumor activity in the combination studies observed in Fig. 2 (as compared to Fig. 1) I find most impressive.

Comments:

Introduction:

- 1. I encourage the authors to include investigational treatments outside of the Roche family of compounds (such as the CEA-TCB). Examples of CD3-based bispecifics that are being tested in non-inflamed tumors, for example, AMG160 and TNB 858 in prostate cancer.***

Thank you for your suggestion. We included a description of PSMA-BiTE and tebentafusp, as recommended.

- 2. ERY974 is mentioned to be evaluated in a clinical trial. In clinicaltrials.gov, ERY974, however, only a single ERY974 study was shown, with a completion date of August 2019. The authors should update this information and disclose where the antibody is currently being clinically investigated.***

Thank you for your comment. As you pointed out, we completed a global Phase I study (NCT02748837) in 2019. Currently, we are conducting two further phase I studies (JapicCTI-194805/NCT05022927). The clinicaltrials.gov website mainly lists clinical trials conducted in the USA. Therefore, the trial information for our study JapicCTI-194805 (ERY974 monotherapy with tocilizumab premedication) is not included there. Furthermore, we have recently started a new global phase I study (NCT05022927) to evaluate the combination of ERY974 with atezolizumab and bevacizumab, which is listed in clinicaltrials.gov. I corrected in the main text.

3. Line 70 indicates weekly injection is sufficient to maintain the antibody at sufficient levels in the blood. The manuscript referred to, however, only shows a mouse study. Where is the conclusion based on? Is ref 12 not more relevant in this context. What is known about PK in humans from a completed clinical study?

We appreciate that you pointed out that we referenced the wrong literature. We have corrected this in the revised manuscript. Our previous data showed that the $t_{1/2}$ of ERY974 in cynomolgus monkeys is approximately 3.3 days¹. Using these data, we estimated the human PK of ERY974 using single-animal species allometric scaling methods. We estimated that the $t_{1/2}$ of ERY974 in humans is approximately 5.1 days, by which we judged that weekly administration would be sufficient to maintain ERY974 levels in blood at the target concentration without accumulation. We have attached the simulation data (our internal data) of the PK profile of ERY974 as a reference.

In fact, we confirmed that weekly dosing achieved the target concentration of ERY974 (more than 2 ng/mL) in blood after the 3rd or 5th infusion from our phase I study (NCT02748837), as expected¹⁰. In short, the target concentration of ERY974 in humans was calculated from the EC₅₀ of TDCC (our internal unpublished data).

Results:

- 4. *Figure 1 b. It is not clear if the black dots show the tumor data in the presence of T cells. Confirm that T cells were injected in both, and the differences observed were only due to the presence or absence of the bispecific T cell engager.***

Yes, the black dots show the tumor size data in the presence of T cells. Therefore, the difference in tumor size between the ERY974 and vehicle groups was described with and without ERY974 administration.

- 5. *Did the authors study tumor growth in the absence of T cells as well? Alloreactivity is often challenging in these types of studies. How was this controlled for? Were donors preselected to avoid alloreactivity against the cellines shown?***

You have asked an important question. As stated by you, alloreactivity is very important for evaluating the efficacy of TRAB in a murine model. Normally, we do not use matched HLA between effector cells and tumor cells because it is very difficult to find specific PBMCs with matched HLA with target cell lines. Furthermore, in most cases, the HLA type of the commercially available human cell lines is not disclosed. Hence, we were obliged to not take into account HLA type, despite our initial concerns regarding alloreactivity. However, so far, clear alloreactivity has not been observed in our preclinical model (i.e., both the huNOG and T cell-injected models). We added the data representing that NCI-H446 tumor growth in mice administered with PBS + T cells is comparable to that in mice administered with only PBS (Fig. S1c). In the literature for preclinical studies of other TRABs, similar models to ours (e.g., T cell injected model, humanized mouse model) have been used. In CEA-TCB¹⁹, PSMA-BiTE¹³, or DLL3-BiTE¹¹, no description regarding HLA matching between effector cells and cancer cells is found in the Methods section of the manuscripts, presumably because they did not use matched HLAs between effector cells and tumor cells.

- 6. *In light of the above, were the in vivo studies performed with T cells derived from multiple donors with similar results. In other words, can the authors exclude that differences in the ‘inflammatory’ or ‘non-inflammatory’ phenotype observed is a result of alloreactivity of the T cells rather than a characteristic of these xenografts. This information should be added as supplementary data.***

This is also an important query. We used T cells from different donors and obtained similar data several times. Per your comment, we have added the efficacy data of ERY974 against PC10 and NCI-H446 in different huNOG lots in Supplementary Fig. S1a-b. The TGI of ERY974 in PC10/huNOG “lot A” and PC10/huNOG “lot B” were 104% and 105%, respectively. The TGI of ERY974 in NCI-H446/huNOG “lot C” and NCI-H446/huNOG “lot D” were 38% and 49%, respectively. These results suggest that tumor characteristics are more important in deciding the TME and thus, alloreactivity may not need to be considered.

7. NCI-H446 tumors were treated with a combination of ERY974, paclitaxel, and cisplatin. MKN45 tumors were treated with a combination of ERY974 and capecitabine. It is unclear how these tumors and treatments were matched. Were NCI-H446 not treated with capecitabine and vice versa MKN45 not treated with paclitaxel or cisplatin / ERY974 combi? Or did these treatments not give the desired results. Please elaborate on the rationale of treatment choice and discuss the results from combination treatments with other tumors. It would be critical to demonstrate that the combis work more generally and not just in one tumor model.

Thank you for your comment. We consider hepatocellular carcinoma as the target cancer type for ERY974, followed by lung and gastric cancers, as GPC3 is highly expressed in these types of cancers¹. In Fig. 2, we prioritized matched combinations of chemotherapy drugs and tumor types where these chemotherapy drugs are clinically used. For example, we combined ERY974 with paclitaxel or cisplatin in the NCI-H446 (lung) tumor model because cisplatin and/or paclitaxel are frequently used as standard therapies for lung cancer², especially in patients with gene alterations negative for *EGFR*, *ALK*, *ROS1*, *BRAF*, *MET*, or *RET*, etc, and we combined ERY974 with capecitabine in the MKN45 (gastric) tumor model because fluorouracil-related drugs including capecitabine are standard therapies for gastric cancer³. However, based on your suggestion, we have added the data for the combination of ERY974 with paclitaxel or cisplatin in MKN45 and the combination of ERY974 with capecitabine in NCI-H446 (Supplementary Fig.3). It should be noted that cisplatin is sometimes used in gastric cancer treatment in combination with fluorouracil-related drugs as first-line treatment, while paclitaxel is also used in gastric cancer as second-line treatment. In most cases in our efficacy studies, the combination of ERY974 with chemotherapy drugs tended to increase efficacy.

Discussion:

8. Line 268 says that IFN and TNF convert capecitabine. However, I believe that this effect occurs indirectly through the induction of thymidine phosphorylase expression by the tumor. Please correct. It would be interesting to discuss how general a phenomenon might be, or could this be restricted to the MKN45 model? Also see my last comment on the results section above.

Thank you for your comment. Yes, you are right. We changed the sentences to “Overall, these data show that IFN γ and TNF α secreted from T cells during TDCC can induce *TYMP* gene transcription followed by TP protein synthesis, which efficiently converts capecitabine to 5'-FU at the tumor site.”

As for the phenomenon of TP induction by cytokines, we tested whether TP is induced by IFN γ or TNF α in other cell lines (e.g., NCI-H446, PC10, and MKN74). We confirmed that IFN γ and TNF α can induce *TYMP* gene expression in all the cell lines tested, although TP protein expression could not be detected in NCI-H446 by western blotting, as basal expression levels of TP were too low (Fig. S9c-d). These results demonstrate that our findings are not restricted to the MKN45 cell line.

Comments from Reviewer #4 (Remarks to the Author): with expertise in cancer and transcriptomic

In the study “Combination of the T cell-redirecting bispecific antibody ERY974 and chemotherapy reciprocally enhances antitumor activity against non-inflamed tumors” Sano et al. describe their investigation of combination effects of the T cell-redirecting bispecific antibody ERY974 and chemotherapy in non-inflamed tumors using a combination of techniques in cell lines and mouse xenograft models.

They first demonstrate the evidence that high expression of GPC3 is not sufficient to induce tumor regression, as evident by limited effect in NCI-H446 xenografts. Surprisingly, they also demonstrate that high expression of GPC3 is also not necessary for an ERY974 anti-tumor effect, as evident by the tumor growth inhibition in MKN74 xenograft, where MKN74 has low expression of GPC3. They show evidence that immune cell infiltration is predictive for the efficacy of ERY974 monotherapy.

Question:

- 1. Authors speculate that “that residential myeloid cells may help to increase number of T cells after ERY974 treatment by secreting cytokines, or chemokines. “As according to methods, RNA material was collected from MKN74 xenograft treated with ERY974, I wonder whether it would be possible to check if the author’s cytokine mediated T-cell recruitment speculation can be supported by data, by performing RNA-seq of those samples.*

Thank you for providing an interesting suggestion. Per your suggestion, we examined the gene expression of chemokines involved in the recruitment of T cells in MKN74 tumors. We found that the gene expression of representative T cell chemokines such as CCL3, CCL4, CCL5, CXCL9, CXCL10, and CXCL11²⁰ at baseline (Vehicle group) in MKN74 tumors tends to be comparable to that in PC10. Furthermore, after ERY974 administration, the expression levels of these cytokines became much higher in MKN74 tumors, as well as the highest out of all three tumors (Fig.S1e). These data support our speculation that residual myeloid cells may contribute to T cell recruitment, as described in the Results section.

The authors further demonstrate an increased anti-tumor effects of combination

treatment of ERY974 with chemotherapy in tumors with low infiltration of immune cells (NCI-H446 and MKN45). They further investigate the mechanism of such increased anti-tumor effect. They demonstrate that in NCI-H446 model in a combination treatment group there is an increased evidence of several biomarkers of T-cells cells presence and activity. Through histopathological analysis the authors demonstrated increased presence of immune cells within the tumor in the combination treatment group, suggesting that chemotherapy enhances ERY974 effect by disturbing the tumor structure and allowing more immune cells and more of ERY974 to infiltrate the tumor.

The authors confirmed their finding using RNA from MKN45 xenograft.

The authors also show that not only chemotherapy enhances the effect of ERY974 by allowing more infiltration, but also that ERY974 treatment increases the efficacy of chemotherapy. They show this by RNA analysis of MKN45 xenograft.

Question:

2. If the authors have the data for both MKN45 and NCI-H446 xenografts, why do they only show data for one of the models in the rest of the manuscript?

Thank you for your comment. We consider hepatocellular carcinoma as the target cancer type for ERY974, followed by lung cancer and gastric cancers, as GPC3 is highly expressed in these types of cancers¹. In Fig. 2, we showed the representative data of matched combinations of chemotherapy drugs and tumor types where these chemotherapy drugs are clinically used. For example, we combined ERY974 with paclitaxel or cisplatin in the NCI-H446 (lung) tumor model because cisplatin and/or paclitaxel are frequently used as standard therapies for lung cancer², especially in patients with gene alterations negative for *EGFR*, *ALK*, *ROS1*, *BRAF*, *MET*, or *RET*, etc, and we combined ERY974 with capecitabine in the MKN45 (gastric) tumor model because fluorouracil-related drugs including capecitabine are standard therapies for gastric cancer³. However, based on your suggestion, we have added the data for the combination of ERY974 with paclitaxel or cisplatin in MKN45 and the combination of ERY974 with capecitabin in NCI-H446 (Supplementary Fig.3). It should be noted that cisplatin is sometimes used in gastric cancer treatment in combination with fluorouracil-related drugs as first-line treatment, while paclitaxel is also used in gastric cancer as second-line treatment. In most cases in our efficacy studies, the combination of ERY974 with

chemotherapy drugs tended to increase efficacy.

To confirm their hypothesis about enhanced chemotherapy effect in combination treatment, the authors examine TP expression in all groups using western blotting, and find that TP expression is increased in combination treatment.

Comment:

3. If the data is available, this finding should be confirmed from RNA data.

This is also an important suggestion. Per your comment, we examined the RNAseq data of the MKN45 tumor on day 14 after the administration of ERY974 and capecitabine. As expected, *TYMP* gene expression was significantly higher in the combination group than in the ERY974 monotherapy group (Fig. 7c). These data indicate that TP protein upregulation is preceded by transcriptional induction in combination therapy.

4. Cell line SK-pca13a is not mentioned anywhere in the main text (except figure legend).

Thank you for pointing out. SK-pca13a is engineered to express GPC3 in SK-HEP-1 which does not express GPC3, and used to compare TDCC between unlabeled ERY974 and FITC-labeled ERY974 (Fig. 5a). We have added a description of SK-pca13a in the main text of our manuscript.

5. Though the text the authors identify the time points where RNA was collected, but they do not specify whether this RNA was subjected to sequencing of Nanostring quantification method. It therefore came as a surprise to me to see that both NanoString and RNA-seq were used. Stating more clearly in the text where quantification is done by RNA-seq and where is it done by Nanostring would address this comment.

Thank you for your comment. We consider RNAseq to be more important. So, RNA data were basically obtained by RNAseq (Fig. 3c-e, Fig. 6c-d, Fig.7a, Fig.S1d-e, Fig.S5c-d, Fig.S8a-b). However, we also obtained a few data by nCounter with some reasons. We described three RNA data obtained by nCounter analyses in our manuscript. The first data was to determine the appropriate timing for observing the combination therapy of ERY974 + capecitabine on days 3, 7, and 14 (Fig. 6b). The reason why we used nCounter here is to roughly estimate the best timing to observe the combined effect of ERY974 and

capecitabine, which was determined to be day 14 eventually. Therefore, we conducted RNAseq analysis only for day 14 samples combined with additional samples to obtain expression data for more variety of genes, as shown in Fig. 6c-d. The second data was to show changes of gene expression in combination group of EGFR-TRAB and paclitaxel in MKN45 tumor, that we added in the revised manuscript (Fig.10g). The reason why we used nCounter for this analysis is just to save time as nCounter data can be obtained in shorter time than RNAseq data.

The third data using nCounter was to support the synergistic cell growth inhibition effect for the combination of IFN γ + 5'-DFUR (5'-FU precursor and direct substrate of TP) *in vitro* (Fig. S9e). Gene signature analysis is available in the nSolver4 advanced analysis software. We utilized pathway analysis in the nCounter data to assess the synergistic cell growth inhibition effect for the combination of IFN γ + 5'-DFUR. Per your suggestion, we have described in the main text and relevant figure legends which RNA analysis was used.

6. *In the methods section for transcriptome analysis it is not indicated which software was use to align reads to mouse and human genome. Ideally, the authors should provide the name and version of the software used, as well as the list of parameters that were used for the analysis (if different from default).*

We used bowtie-1.1.2 to align the reads to the mouse and human genomes. We used the RSEM parameters as “-n 2 -e 99999999 -l 25 -I 1 -S -X 1000 -a -m 200”. We added this description in Method section.

7. *The sentence describing normalization procedure for RNA-seq reads is confusing. RSEM provides transcripts per million (TPM), which is a currently accepted normalization method, and fragments per kilobase per million (FPKM) is calculated using a different procedure and tool, and is not advised to be used. From the sentence, it is not clear what was done with the data. Please state clearly whether the TPM or FPKM was used. From this sentence, it follows that FPKM normalization was applied to TPM values, which should not be done.*

We used the FPKM value from the RSEM outputs (sample.genes.results), containing both TPM and FPKM.

<https://deweylab.github.io/RSEM/rsem-calculate-expression.html>

8. *Conclusion: I found the study to be very interesting, and overall of good quality (given the comments above are addressed). The evidence of increased anti-tumor effects of combination treatment in mouse models is convincing, though the major drawback in my opinion is use of a single cell line as evidence of the mode of action (the data seems to be available though, but it is not evident from the manuscript, as single cell line is used for each point made by the authors). I am still intrigued by the effect of ERY974 monotherapy MKN74 xenograft, but this seems to be out of scope of this study, as the main focus is the combination therapy and it's mode of action.*

My background is in Cancer genomics and bioinformatics analysis, with extensive experience in transcriptomics. I have worked with RNA-seq data and Nanostring data, from human cell lines, mouse models and cancer patients. Therefore my questions might not require manuscript modification and may be addressed by the authors' explanation, but my comments should be addressed in the manuscript.

*Sincerely,
T. Aneichyk*

Thank you for your encouraging comments. As described above (see the section of response to your question 2), we have included data on the combinations of ERY974 with paclitaxel, cisplatin, or capecitabine in MKN45 tumor (Fig. S3a-b, Fig. 2e-f) and the combination of ERY974 with paclitaxel, cisplatin, or capecitabine in NCI-H446 tumor (Fig. 2a-b, Fig. 2c-d, and Fig. S3c). In general, we observed a similar synergistic antitumor effect in all combinations. As for MKN74 model, we are also interested in the function of myeloid cells at baseline for making in-flamed TME. Per your suggestion, we added some data to support our hypothesis as described above.

References

1. Ishiguro T, et al. An anti-glypican 3/CD3 bispecific T cell-redirecting antibody for treatment of solid tumors. *Science translational medicine* 9, (2017).
2. NCCN.org. NCCN Clinical Practice Guidelines in Oncology Non-Small Cell Lung Cancer.) (2021).
- 3 NCCN.org. NCCN Clinical Practice Guidelines in Oncology Gastric Cancer. (2021).
4. Waaijer SJ, et al. Preclinical PET imaging of bispecific antibody ERY974 targeting CD3 and glypican 3 reveals that tumor uptake correlates to T cell infiltrate. *Journal for immunotherapy of cancer* 8, (2020).
5. Socinski MA, et al. Atezolizumab for First-Line Treatment of Metastatic Nonsquamous NSCLC. *The New England journal of medicine* 378, 2288-2301 (2018).
6. Gandhi L, et al. Pembrolizumab plus Chemotherapy in Metastatic Non-Small-Cell Lung Cancer. *The New England journal of medicine* 378, 2078-2092 (2018).
7. Cortes J, et al. Pembrolizumab plus chemotherapy versus placebo plus chemotherapy for previously untreated locally recurrent inoperable or metastatic triple-negative breast cancer (KEYNOTE-355): a randomised, placebo-controlled, double-blind, phase 3 clinical trial. *Lancet* 396, 1817-1828 (2020).
8. Boudousquie C, Bossi G, Hurst JM, Rygiel KA, Jakobsen BK, Hassan NJ. Polyfunctional response by ImmTAC (IMCgp100) redirected CD8(+) and CD4(+) T cells. *Immunology* 152, 425-438 (2017).
9. Middleton MR, et al. Tebentafusp, A TCR/Anti-CD3 Bispecific Fusion Protein Targeting gp100, Potently Activated Antitumor Immune Responses in Patients with Metastatic Melanoma. *Clinical cancer research : an official journal of the American Association for Cancer Research* 26, 5869-5878 (2020).

10. Safran H, et al. Abstract CT111: Results of a phase 1 dose escalation study of ERY974, an anti-glypican 3 (GPC3)/CD3 bispecific antibody, in patients with advanced solid tumors. *Cancer research* 81, CT111-CT111 (2021).
11. Giffin MJ, et al. AMG 757, a Half-Life Extended, DLL3-Targeted Bispecific T-Cell Engager, Shows High Potency and Sensitivity in Preclinical Models of Small-Cell Lung Cancer. *Clinical cancer research : an official journal of the American Association for Cancer Research* 27, 1526-1537 (2021).
12. Owonikoko TK, et al. Phase I study of AMG 757, a half-life extended bispecific T-cell engager (HLE BiTE immune therapy) targeting DLL3, in patients with small cell lung cancer (SCLC). *Journal of Clinical Oncology* 38, TPS9080-TPS9080 (2020).
13. Deegen P, et al. The PSMA-targeting Half-life Extended BiTE Therapy AMG 160 has Potent Antitumor Activity in Preclinical Models of Metastatic Castration-resistant Prostate Cancer. *Clinical cancer research : an official journal of the American Association for Cancer Research* 27, 2928-2937 (2021).
14. Tran B, et al. 609O Results from a phase I study of AMG 160, a half-life extended (HLE), PSMA-targeted, bispecific T-cell engager (BiTE®) immune therapy for metastatic castration-resistant prostate cancer (mCRPC). *Annals of Oncology* 31, S507 (2020).
15. B. Tran¹ LH, T. Dorff³, M. Rettig⁴, M.P. Lolkema⁵, J. Machiels⁶, S. Rottey⁷, K. Autio⁸, R. Greil⁹, N. Adra¹⁰, C. Lemech¹¹, M. Minocha¹², F. Cheng¹³, H. Kouros-Mehr¹⁴, K. Fizazi¹⁵. Results from a phase I study of AMG 160, a half-life extended (HLE), PSMA-targeted, bispecific T-cell engager (BiTE®) immune therapy for metastatic castration-resistant prostate cancer (mCRPC). In: *ESMO Virtual Congress*) (2020).
16. Ströhlein MA, Heiss MM. The trifunctional antibody catumaxomab in treatment of malignant ascites and peritoneal carcinomatosis. *Future Oncol* 6, 1387-1394 (2010).
17. Borlak J, Langer F, Spanel R, Schondorfer G, Dittrich C. Immune-mediated liver

injury of the cancer therapeutic antibody catumaxomab targeting EpCAM, CD3 and Fcγ receptors. *Oncotarget* 7, 28059-28074 (2016).

18. Galluzzi L, Buqué A, Kepp O, Zitvogel L, Kroemer G. Immunogenic cell death in cancer and infectious disease. *Nat Rev Immunol* 17, 97-111 (2017).
19. Bacac M, et al. A Novel Carcinoembryonic Antigen T-Cell Bispecific Antibody (CEA TCB) for the Treatment of Solid Tumors. *Clinical cancer research : an official journal of the American Association for Cancer Research* 22, 3286-3297 (2016).
20. Yao X, Matosevic S. Chemokine networks modulating natural killer cell trafficking to solid tumors. *Cytokine Growth Factor Rev* 59, 36-45 (2021).

Reviewers' Comments:

Reviewer #1:

Remarks to the Author:

The authors have addressed major and minor comments from the initial submission. However, a handful of concerns still remain:

Major:

1) In the initial review, outlined in Major 1-1, the concern was raised about not showing survival benefit for either therapeutic groups from the experiments conducted. The authors explain that death as an endpoint is not preferred due to IACUC regulations, which is reasonable and is indeed the general approach in the field. Euthanasia based on predefined criteria (including tumor burden as the authors mention) is a well-accepted way to generate Kaplan-Meier survival estimates and should be included in the study. It seems that the authors have this data and would be critical to include in the figures.

2) In figure 2, the termination of tumor volume curve does not seem to correspond to a specific tumor volume. For example, Fig 2a ERY974 group is terminated at a much lower tumor burden than the paclitaxel group or in Fig 2d, ERY974 curve ends at the similar tumor burden as paclitaxel, while this latter group continues to be alive with growing tumors. Why such difference in the timing of euthanasia? Were the ERY974 treated mice die of non-tumor related reasons?

3) The authors refer to a technical issue for not performing the T cell tracking experiment which could explain the differences in T cell infiltration. These experiments would have given important mechanistic insight which is currently lacking.

4) The authors responded to the Comment Major 4, but may have misunderstand the question. The RNAsequencing results show upregulation of genes related to a wide range of programs: naïve, inhibitory, cytokines, effector, costim, transcription factors and T reg. All of these programs cannot be overexpressed at the same time. T cells downregulate the naïve program upon antigen encounter and start to different eventually to fully functional effectors. What can explain the increase in genes related to naïve T cells and effector T cells at the same time? Could these findings be related to technicality of having more T cells from the combination group?

Minor: Consider adding the tumor cell lines as titles to each panel (or rows) in Fig 2 for easier interpretation of the data.

Reviewer #3:

Remarks to the Author:

I thank the authors for their detailed responses and providing a revision. The modifications and additions have further improved the papers. The authors satisfactorily answered my queries from my perspective.

Reviewer #5:

Remarks to the Author:

The authors have adequately addressed raised issues.

The authors stated now clearly which methods and normalization for gene expression analyses they have used. However, I recommend to use TPM instead of FPKM. In RSEM, FPKM is derived from TPM what should be not done -as also stated in the previous comments - although this should not have an high impact on the pronounced results and conclusions.

Since the authors have added information on xenograft models with different tumor cell lines treated with other therapy combinations find this highly intersting manuscript has substantially improved, where they studied combination treatment of non-inflamed tumors in mouse tumor models by CD3xGPC3 bispecific antibody and chemotherapeutic drugs.

Dear Reviewer 1,

We would like to thank you for the opportunity to respond to your valuable comments and queries on our manuscript, “Combination of T cell-redirecting bispecific antibody ERY974 and chemotherapy reciprocally enhances efficacy against non-inflamed tumours.” We also appreciate the time and effort you have dedicated to providing insightful feedback on ways to strengthen our paper. We have provided response to each of your comments and queries and have addressed any relevant issues. We hope that the responses below satisfactorily address all the issues and concerns you have raised.

Each of your comments is written in bold and is followed by a point-by-point response to each question or comment. The responses have been developed in consultation with all coauthors.

Reviewer # 1 (Remarks to the Author):

The authors have addressed major and minor comments from the initial submission. However, a handful of concerns still remain:

Major:

1) In the initial review, outlined in Major 1-1, the concern was raised about not showing survival benefit for either therapeutic groups from the experiments conducted. The authors explain that death as an endpoint is not preferred due to IACUC regulations, which is reasonable and is indeed the general approach in the field. Euthanasia based on predefined criteria (including tumor burden as the authors mention) is a well-accepted way to generate Kaplan-Meier survival estimates and should be included in the study. It seems that the authors have this data and would be critical to include in the figures.

Thank you for your comments. According to your suggestion, we have generated Kaplan–Meier survival curves (displayed below), using huNOG data (Fig. 2b, 2d, and 2f). However, there are some issues. The IACUC criteria define the endpoint as a tumour size of 2,000 mm³ as described in "In vivo efficacy study using the humanized mouse model" subsection of the Methods section We normally sacrifice all the mice belonging to the group if the tumour size of one mouse in the group exceeds 2,000 mm³, even if the tumour sizes of the other mice are less than 2,000 mm³. As we were

supposed to compare the mean tumour volume among groups ($n = 5$), the tumour size measurement of the remaining mice could be misleading. Therefore, in some cases, we only know the 80% survival timing (when only one mouse's tumour size exceeded 2000 mm^3 and all the mice in that group were sacrificed at that time). Then we set the endpoint as 1,000 mm^3 , a much shorter tumour size for the combination of ERY974 with paclitaxel or cisplatin (Fig. 2b and 2d). However, we still observed only 80% survival timing for paclitaxel monotherapy group. When we combine ERY974 with capecitabine (Kaplan–Meier survival curves of Fig. 2f), we set the endpoint as a tumour size of 500 mm^3 . In this study, we completed tumour size measurements before any of the tumours reached 2,000 mm^3 because we could determine that a synergistic effect was apparent within a shorter time frame. For these reasons, our experimental design is not suitable to generate a Kaplan–Meier survival curve. These data are just for reference and may not be suitable for inclusion in the manuscript or supplementary figures.

2) In figure 2, the termination of tumor volume curve does not seem to correspond to a specific tumor volume. For example, Fig 2a ERY974 group is terminated at a much

lower tumor burden than the paclitaxel group or in Fig 2d, ERY974 curve ends at the similar tumor burden as paclitaxel, while this latter group continues to be alive with growing tumors. Why such difference in the timing of euthanasia? Were the ERY974 treated mice die of non-tumor related reasons?

Thank you for your query. We determined the termination point of each group in the study according to the IACUC regulations. The most important criterion is tumour size. In general, when the tumour size of at least one of the mice in each group was more than 2,000 mm³ and that mouse is sacrificed, we completed the tumour size measurement of that group. Normally, our plan is to compare the mean tumour size of each group. However, we sometimes terminate a study before the tumour size reaches 2,000 mm³ when we can judge the effect of drugs in a shorter time or when huNOG mice are used. The huNOG mice sometimes develop graft versus host disease (GvHD) as they age due to implantation of human CD34+ cells. Therefore, we carefully observe the condition of the mice during the study and determine a suitable termination point. Regarding the tumour size in the ERY974 group (Fig. 2a), we completed the tumour size measurement on day 37 because the tumour size of one mouse in this group reached 2,165 mm³. On day 37, the mean tumour size of this group (n = 5) was 813 mm³. Regarding the paclitaxel group, the tumour in one mouse rapidly grew to over 3,000 mm³ from 1,833 mm³ in size on day 62, because of which we terminated tumour size measurement; the average tumour size in this group on day 62 was 1,879 mm³. The tumour sizes at the termination point in the ERY974 (on day 37) and paclitaxel groups (on day 62) were 148, 2165, 725, 916, and 112 mm³ (average 813 mm³), and 1992, 1600, 3303, 1459, and 1041 mm³ (average 1,879 mm³), respectively. The tumour size over 3,000 mm³ caused the higher mean value in the paclitaxel group. With respect to the tumour sizes in the ERY974 group from Fig. 2d, on day 39 when tumour size measurement was completed, the average tumour size was 1,015 mm³, which was not much different from that on day 37 (813 mm³) in Fig 2a. However, in our experimental design, we understand that the termination point in each group is largely affected by an individual difference, and generally, individual differences of immunotherapy drugs are higher. Therefore, the observation period may not be consistent between studies.

3) The authors refer to a technical issue for not performing the T cell tracking experiment which could explain the differences in T cell infiltration. These experiments would have given important mechanistic insight which is currently

lacking.

Thank you for your suggestion. I believe that you are raising an important point. Through this comment, we have reconsidered what happened to the T cells after administration of the chemotherapy drug. According to the literature, the blood concentration of 20 mg/kg paclitaxel in mice, which is the same dose used in our studies, is 120 $\mu\text{g/mL}$ at C_{max} and reduces to approximately to 30-40 ng/mL at 18 h¹. In the huNOG model, human T cells that are already present, may be susceptible to chemotherapy, which may inhibit the efficacy of ERY974. The C_{max} level of paclitaxel (120 $\mu\text{g/mL}$) could likely result in cytotoxicity in most T cells. However, we have not observed any evidence that the combination of chemotherapy with ERY974 results in decreased efficacy in any of the models tested. Human T cells in huNOG mice are present in many tissues as well as in the blood, spleen, thymus, and bone marrow². In our studies, we also observed the distribution of human T cells in the tumour at baseline in a PC10 xenograft model (Fig. S2a). We speculate that these tissue-resident T cells might be protected by the structure of tissues and exempted from attack of paclitaxel. Furthermore, the paclitaxel concentration in tissues might be lower than that in blood. As for the T cell-injected model, paclitaxel was administered one day before the administration of *ex-vivo* expanded T cells and ERY974. As described above¹, on the day following administration (18h), the paclitaxel concentration in the blood is approximately 30-40 ng/mL. In our unpublished data, we found that PBMCs pretreated with 1 μM paclitaxel (which corresponds to 854 ng/mL) for 24 h has a less impact on TDCC by ERY974 (described below). Therefore, administered human T cells on the next day may not be seriously affected by palitaxel even if all of them remain in the blood. In our previous study, ⁸⁹Zr-labeled ERY974 was detected in the tumour through its GPC3 arm and spleen and lymph nodes through its CD3 arm in the huNOG model³. Thus, human T cells are present in lymphoid tissues, and ERY974 can recruit fresh T cells from lymphoid tissues to the tumour site. Of course, it cannot be denied that paclitaxel has an impact on T cell proliferation. However, we believe that initial T cell damage from chemotherapy can be overcome by ERY974, which strongly induces proliferation of T cells. In our murine models, a balance between chemotherapy-induced cytotoxicity against human T cells and ERY974-induced proliferation of human T cells might be more favorable to the effect of ERY974. To examine if this hypothesis is reasonable, a T cell-labeling study would be useful to clarify the fate of T cells, as you suggested.

xCELLigence assay

Effector cell: PBMC (E/T 20)

Pretreated with or without paclitaxel (0.01, 1 µM) for 24h

Target cell: PC10

Reaction time: 72h

IC50 of ERY974:

Control (w/o paclitaxel): 0.0859 µg/mL

With 0.01 µM paclitaxel: 0.0953 µg/mL

With 1 µM paclitaxel: 0.0991 µg/mL

4) The authors responded to the Comment Major 4, but may have misunderstand the question. The RNAsequencing results show upregulation of genes related to a wide range of programs: naïve, inhibitory, cytokines, effector, costim, transcription factors and T reg. All of these programs cannot be overexpressed at the same time. T cells downregulate the naïve program upon antigen encounter and start to different eventually to fully functional effectors. What can explain the increase in genes related to naïve T cells and effector T cells at the same time? Could these findings be related to technicality of having more T cells from the combination group?

Thank you for the explanation. I apologize for not explaining my response clearly enough. As you stated, naïve T cells are primed by dendritic cells (DCs) through interaction of the TCR and human leukocyte antigen (HLA) complex and cancer antigens, which induces differentiation to cytotoxic T cells. This is the sequential process since the activation of naïve T cells requires antigen encounter as the first step. However, T cell redirecting antibodies (TRABs), including ERY974, might skip this process and directly activate various T cells (e.g., naïve, cytotoxic, and memory T cells) by inducing agonistic signal in T cells. We believe that TRABs might not have preference for subtypes of T cells if they express CD3. According to this hypothesis, we believe that it is possible that naïve T cells and cytotoxic T cells are simultaneously activated by TRABs. Consistent with this, it was recently reported that blinatumomab, a bi-specific T-cell engager (BiTE) antibody targeting CD19, activated CD8⁺ effector memory T cells, CD4⁺ central memory T cells, naïve T cells, and regulatory T cells (Tregs) at the same time in a single cell RNA analysis

*in vitro*⁴. Currently, we are conducting a similar single cell RNA experiment using ERY974 (data not shown). Preliminary data suggest that ERY974 treatment causes enrichment of various T cell subtypes, including naïve and cytotoxic T cells. However, it is of note that only a few subtypes in naïve, cytotoxic, and memory T cells are activated. Our data and those of others might support our hypothesis. Furthermore, indirect activation is also possible. First, ERY974 induced cytokine expression in T cells, which might activate various subtypes of T cells. This indirect activation caused simultaneous activation of naïve and cytotoxic T cells. For example, in our previous study, we confirmed that ERY974 induced IL2 secretion during TDCC⁵. IL2 is a potent inducer of proliferation for most T cells irrespective of subtypes.

Minor: Consider adding the tumor cell lines as titles to each panel (or rows) in Fig 2 for easier interpretation of the data.

Thank you for your comment. I added the tumour cell lines and models in each panel in Figure 2a-f, Figure 8e-f, Supplementary Figure 1a-c, and Supplementary Figure 3a-c.

Reviewer #3 (Remarks to the Author):

I thank the authors for their detailed responses and providing a revision. The modifications and additions have further improved the papers. The authors satisfactorily answered my queries from my perspective.

Reviewer #5 (Remarks to the Author):

The authors have adequately addressed raised issues.

The authors stated now clearly which methods and normalization for gene expression analyses they have used. However, I recommend to use TPM instead of FPKM. In RSEM, FPKM is derived from TPM what should be not done -as also stated in the previous comments - although this should not have an high impact on the pronounced results and conclusions.

Since the authors have added information on xenograft models with different tumor

cell lines treated with other therapy combinations find this highly interesting manuscript has substantially improved, where they studied combination treatment of non-inflamed tumors in mouse tumor models by CD3xGPC3 bispecific antibody and chemotherapeutic drugs.

Thank you once again for allowing us to improve our manuscript with your valuable comments and queries.

Sincerely,

Yuji Sano, Ph.D.

Discovery Pharmacology Department

200 Kajiwara, Kamakura, Kanagawa 247-8530, Japan

Tel: +81-467-47-6241

Fax: +81-467-47-2234

Email: sanoyuj@chugai-pharm.co.jp

References

1. Sparreboom, A., van Tellingen, O., Nooijen, W.J. & Beijnen, J.H. Nonlinear pharmacokinetics of paclitaxel in mice results from the pharmaceutical vehicle Cremophor EL. *Cancer Research* **56**, 2112-2115 (1996).
2. Watanabe, S., *et al.* Hematopoietic stem cell-engrafted NOD/SCID/IL2Rgamma null mice develop human lymphoid systems and induce long-lasting HIV-1 infection with specific humoral immune responses. *Blood* **109**, 212-218 (2007).
3. Waaijer, S.J., *et al.* Preclinical PET imaging of bispecific antibody ERY974 targeting CD3 and glypican 3 reveals that tumor uptake correlates to T cell infiltrate. *Journal for Immunotherapy of Cancer* **8**, (2020).
4. Huo, Y., *et al.* Blinatumomab-induced T cell activation at single cell transcriptome resolution. *BMC Genomics* **22**, 145 (2021).
5. Harada A, *et al.* In vitro toxicological support to establish specification limit for anti-CD3 monospecific impurity in a bispecific T cell engager drug, ERY974. *Toxicol In Vitro* **66**, 104841 (2020).

Reviewer #1 (Remarks to the Author):

1. The authors provide partial explanation for mouse handling. Each mouse could be followed until their tumor reaches a predefined sacrifice requiring sacrifice. It is not standard of practice to sacrifice the whole group based on a single mouse. Visualizing individual mouse and their tumor burden over time is feasible using spaghetti plots and median time to a given tumor size would more accurately identify the differences between groups. I highly recommend this approach for consideration in the future.
2. The variability of observation periods should have been explained in the Discussion as a potential limitation of the studies.
3. It seems that authors agree with the reviewer's comment. Was the experiment performed? There doesn't seem to be revisions in the manuscript body to explain the current limitations with the T cell trafficking component.
4. The authors provide rationale for the RNAseq findings. However, their additional interpretation of RNAseq results, specifically for the increase in genes expressed across mpx differential and activation state seems to be still missing from the manuscript body.

12th May 2022

Dear Reviewer 1,

We would like to thank you for the opportunity to respond to your valuable comments and queries on our manuscript, “Combination of T cell-redirecting bispecific antibody ERY974 and chemotherapy reciprocally enhances efficacy against non-inflamed tumours” We also appreciate the time and effort you have dedicated to providing insightful feedback on ways to strengthen our paper. We have addressed each of your comments and queries. We hope that responses we provide below satisfactorily address all the issues and concerns you have noted.

Your comments are written in bold followed by a point-by-point response to each question or comment. The revision has been developed in consultation with all coauthors, and each author has approved the final form of this revision.

Reviewer #1 (Remarks to the Author):

1. The authors provide partial explanation for mouse handling. Each mouse could be followed until their tumor reaches a predefined sacrifice requiring sacrifice. It is not standard of practice to sacrifice the whole group based on a single mouse. Visualizing individual mouse and their tumor burden over time is feasible using spaghetti plots and median time to a given tumor size would more accurately identify the differences between groups. I highly recommend this approach for consideration in the future.

Thank you for your comments. So far we have mainly prioritized the % of tumour growth inhibition (TGI) as read-out in our previous studies. However, according to your suggestion, we will make an in vivo study plan so that we could draw graphs showing both TGI and survival curve.

2. The variability of observation periods should have been explained in the Discussion as a potential limitation of the studies.

Thank you for your comments. We added the sentence below in the Discussion section. “According to our protocol of this study, we usually completed tumour size measurement of the group where tumour size in one of the mice reaches approximately 2000 mm³,

which caused variability of observation periods among groups, and an unavailability to draw survival curves. This may cause a limitation of profound assessment for antitumour effect of the combination.”

3. It seems that authors agree with the reviewer’s comment. Was the experiment performed? There doesn’t seem to be revisions in the manuscript body to explain the current limitations with the T cell trafficking component.

So far we have not conducted T cell labelling study as technically it is difficult. Therefore, we added the sentence below in the Discussion section. “Exploitation of labelled T cells in T cell injected models might help to further understand the fate of residual T cells for future study.”

4. The authors provide rationale for the RNAseq findings. However, their additional interpretation of RNAseq results, specifically for the increase in genes expressed across mpx differential and activation state seems to be still missing from the manuscript body.

Thank you for your comments. we added the sentence below in the two parts of Discussion section. I believe these two descriptions could cover MOA of ERY974. The first part is described below.

“We speculate that this is because TRAB functions independently of MHC class I and TCR complex formation. When the number of cytotoxic lymphocytes recognizing tumour-derived peptides presented by MHC class I is limited, the TME likely becomes non-inflamed, hindering the efficacy of ICIs. However, ERY974 can utilize any kind of T cell as an effector cell only if CD3 is expressed, thereby inducing the proliferation of a variety of T cells at the tumour site. Furthermore, unlike ICI, ERY974 can exert TDCC (Fig. 5a and 7d) and antitumour efficacy (Fig. 2a–f), utilizing allogeneic effector cells. These observations strongly suggest that TRAB modality can satisfy unmet medical needs by showing efficacy in non-inflamed tumours. “

The second part is described below.

“In our gene expression analysis, ERY974 upregulated a variety of genes whose expression level differed among subsets of T cells, including naïve T cells, cytotoxic T cells, and Tregs (Fig. 3d, 6c. and Supplementary Fig. 4c). These data suggest that ERY974 could simultaneously induce activation and proliferation of various subsets of T cells. Consistent with this, it was recently reported that blinatumomab, a BiTE antibody

targeting CD19, activated CD8+ effector memory T cells, CD4+ central memory T cells, naïve T cells, and Tregs at the same time in single cell RNA analysis in vitro. Similarly, tebentafusp can utilize various T cells, including CD45RA effector memory, central memory, effector memory, and naïve T cells, as effector cells. We believe that TRABs might not have preference for subtypes of T cells only if they express CD3. Additionally, ERY974 may indirectly induce infiltration and activation of various immune cells through cytokines or chemokines secreted from T cells that are initially activated by ERY974.”

Thank you once again for allowing us to improve our manuscript with your valuable comments and queries.

Sincerely,

Yuji Sano, Ph.D.

Discovery Pharmacology Department

200 Kajiwara, Kamakura, Kanagawa 247-8530, Japan

Tel: +81-467-47-6241

Fax: +81-467-47-2234

Email: sanoyuj@chugai-pharm.co.jp

References

1. Sparreboom A, van Tellingen O, Nooijen WJ, Beijnen JH. Nonlinear pharmacokinetics of paclitaxel in mice results from the pharmaceutical vehicle Cremophor EL. *Cancer research* **56**, 2112-2115 (1996).
2. Watanabe S, *et al.* Hematopoietic stem cell-engrafted NOD/SCID/IL2Rgamma null mice develop human lymphoid systems and induce long-lasting HIV-1 infection with specific humoral immune responses. *Blood* **109**, 212-218 (2007).
3. Waaijer SJ, *et al.* Preclinical PET imaging of bispecific antibody ERY974 targeting CD3 and glypican 3 reveals that tumor uptake correlates to T cell infiltrate. *Journal for immunotherapy of cancer* **8**, (2020).
4. Huo Y, *et al.* Blinatumomab-induced T cell activation at single cell transcriptome resolution. *BMC Genomics* **22**, 145 (2021).
5. Ishiguro T, *et al.* An anti-glypican 3/CD3 bispecific T cell-redirecting antibody for treatment of solid tumors. *Science translational medicine* **9**, (2017).